# Accelerated linear algebra compiler for computationally efficient numerical models: Success and potential area of improvement

**Xuzhen He**📵*

Faculty of Engineering and Information Technology, University of Technology Sydney, Ultimo, NSW, Australia

* xuzhen.he@uts.edu.au

## Abstract

The recent dramatic progress in machine learning is partially attributed to the availability of high-performant computers and development tools. The accelerated linear algebra (XLA) compiler is one such tool that automatically optimises array operations (mostly fusion to reduce memory operations) and compiles the optimised operations into high-performant programs specific to target computing platforms. Like machine-learning models, numerical models are often expressed in array operations, and thus their performance can be boosted by XLA. This study is the first of its kind to examine the efficiency of XLA for numerical models, and the efficiency is examined stringently by comparing its performance with that of optimal implementations. Two shared-memory computing platforms are examined–the CPU platform and the GPU platform. To obtain optimal implementations, the computing speed and its optimisation are rigorously studied by considering different workloads and the corresponding computer performance. Two simple equations are found to faithfully modell the computing speed of numerical models with very few easily-measureable parameters. Regarding operation optimisation within XLA, results show that models expressed in low-level operations (e.g., slice, concatenation, and arithmetic operations) are successfully fused while high-level operations (e.g., convolution and roll) are not. Regarding compilation within XLA, results show that for the CPU platform of certain computers and certain simple numerical models on the GPU platform, XLA achieves high efficiency (> 80%) for large problems and acceptable efficiency (10%~80%) for medium-size problems–the gap is from the overhead cost of *Python*. Unsatisfactory performance is found for the CPU platform of other computers (operations are compiled in a non-optimal way) and for high-dimensional complex models for the GPU platform, where each GPU thread in XLA handles 4 (single precision) or 2 (double precision) output elements–hoping to exploit the high-performant instructions that can read/write 4 or 2 floating-point numbers with one instruction. However, these instructions are rarely used in the generated code for complex models and performance is negatively affected. Therefore, flags should be added to control the compilation for these non-optimal scenarios.

**Citation:** He X (2023) Accelerated linear algebra compiler for computationally efficient numerical models: Success and potential area of improvement. PLoS ONE 18(2): e0282265. https://doi.org/10.1371/journal.pone.0282265

**Data Availability Statement:** All relevant data are within the paper and its Supporting Information files.

**Funding:** XH was supported by the Australian Research Council (https://www.arc.gov.au/)

Discovery Early Career Researcher Award (DECRA; DE220100763). The funder had no role in study design, data collection and analysis, decision to publish, or preparation of the manuscript.

**Competing interests:** The author has declared that no competing interests exist.

## 1 Introduction

The pressing problems in many mathematically oriented scientific fields are ubiquitously modelled with partial differential equations, e.g., the elastoplastic deformation of jammed granular materials and their subsequent fluid-like flow after unjamming [1, 2], turbulent air flow around jets [3], the dynamics of financial markets of derivative investment instruments [4], etc. A vast majority of such research relies on numerical models to find approximate solutions to the differential equations and to make reliable predictions. In the last century or so, we have seen significant advances in numerical modelling, including multi-physics coupling with many variables and using fine grids/meshes with massive nodes and cells [2] to achieve realistic simulations. Consequently, the demand for efficiently solving numerical models with many variables and on very fine meshes is increasing.

Traditionally, efficient numerical models are studied in computational complexity theory [5], in which the amount of time, storage, or other resources required to perform numerical simulations are examined theoretically. A classic example is the complexity analysis of the different iterative methods for large systems of linear equations. Analysis [6] shows that if the number of non-zero matrix entries is $N$, and the condition number is $\kappa$, the steepest descent method has a time complexity of $O(\kappa N)$ and the conjugate gradient method has a time complexity of $O(\sqrt{\kappa}N)$, which is thus more efficient. This kind of theoretical analysis is often not enough to model the computing speed of numerical models on modern computers because these computers are all designed with parallel computing capabilities, and efficient implementations must account for the different features of the computers. Consequently, many pieces of research are carried out on the efficient implementation of numerical models on specific computing platforms, including the CPU platform (a shared-memory system with a multi-core central processing unit and the associated main memory) [7, 8], the GPU platform (a shared-memory system with a graphics processing unit and the associated GPU memory) [9], distributed systems [10], and even quantum computing [11, 12].

With the rapid development of new modelling techniques, the demand for rapid prototyping of new numerical models increases in addition to the demand for computationally efficient numerical models. The accelerated linear algebra (XLA) compiler [13] is one of the tools that aim to achieve these (i.e., both computational efficiency and rapid prototyping). The XLA compiler (simply referred to as XLA) takes HLO IR (high-level operation intermediate representation, simply referred to as HLO) as inputs (Fig 1), conducts several rounds of optimisations, and compiles the HLOs into highly efficient computer programs specific to the target computing platform. The optimisation and compilation in XLA happened automatically such that the modellers effortlessly get efficient programs, and they do not need to know the optimisation detail. The HLO inputs to XLA are a board category of array operations and thus most *Python* packages supporting array programming are XLA frontends (e.g., *Tensorflow*, *JAX*, *PyTorch*, etc.) [13]. The target-independent optimisation includes common subexpression elimination, operation fusion, and buffer analysis of memory allocation [13]. Target-dependent optimisation is conducted by considering target-specific features. The target computing platforms for XLA include the x64 CPU platform and the GPU platform (*NVIDIA* GPUs) in the source tree, and new backends can always be added [13]. For the CPU and GPU backends, the *LLVM* compiler [14] is used to compile the *LLVM* intermediate representation into efficient computer programs.

XLA was originally designed to boost the performance of machine-learning models, and performance gain was widely demonstrated. For example, *Google*'s submissions to *MLPerf* (an industry standard for measuring machine learning performance) [15] demonstrate a seven-fold performance gain on the training throughput of XLA-boosted BERT (a transformer-

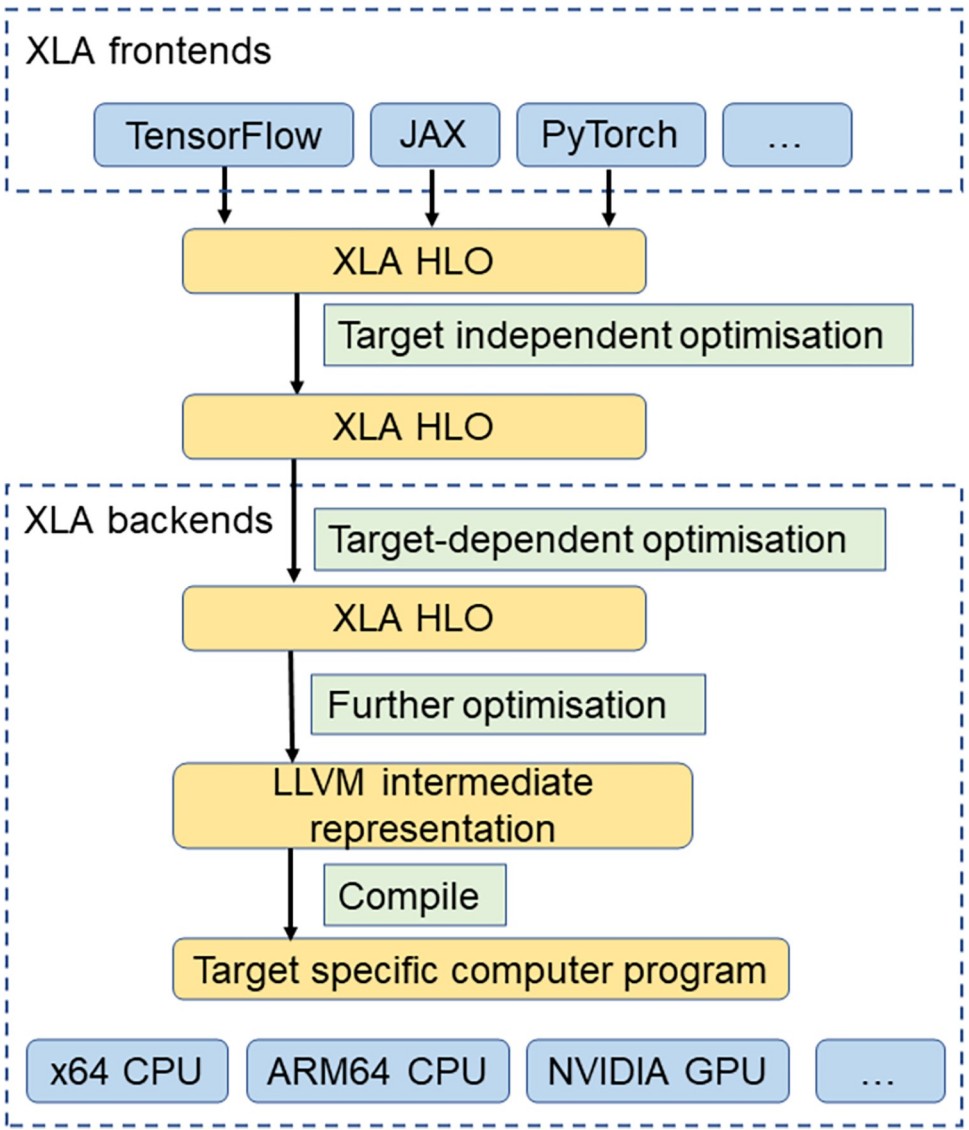

**Fig 1. Architecture of accelerated linear algebra (XLA) compiler.**

based model for natural language processing). Chadha and Siddagangaiah [16] conducted tests on several different models (e.g., two-layer convolutional network, saxpy, matrix multiplication, softmax, and long short-term memory), and found that XLA successfully conducted optimisations for some models, but areas for further improvements were also identified. XLA has also been used to boost the performance of other scientific computing. For example, Lewis *et. al.* [17] demonstrated the efficiency of XLA in matrix multiplication, QR factorisation, linear solution, and application of matrix functions on tensor processing units (TPUs). Sankaran *et. al.* [18] examined the performance of XLA in conducting linear algebra optimisations. Nevertheless, there has been little if any research in the literature about the efficiency of XLA for numerical models.

This study is the first of its kind to examine the efficiency of XLA for a general category of numerical models and the efficiency is examined in a stringently way by comparing the performance of numerical models implemented using XLA on a computing platform with the

maximum possible performance achieved on that computing platform. This study is not meant to be comprehensive, so numerical models defined on regular grids are mainly examined. The scope of the numerical models and some examined examples are explained in Section 2, which belong to two categories (e.g., element-wise operations and finite difference models). *JAX* [19] is used as the XLA frontend and two widely used computing platforms (backends) are examined–the CPU platform and the GPU platform. To obtain the maximum performance on these computing platforms, the computing speed of the numerical models is rigorously studied in Section 4 by considering the different workloads and the corresponding computer performance. Optimal implementations are explored in Section 5, and the computing speed of element-wise operations and finite-difference models are found faithfully modelled by two simple equations with very few easily-measurable parameters. The efficiency of XLA is examined by an index of relative efficiency–the ratio of effective bandwidth between XLA implementations and optimal implementations. XLA is found efficient for some numerical models and for some computing platforms but is not for others, and potential areas of improvement are suggested for the non-optimal scenarios.

## 2 Array programming of numerical models

XLA takes array operations defined in HLO as inputs. An array (or tensor) is a collection of homogenous elements. Matrices are a special case of 2D arrays and vectors 1D arrays. Fig 2 illustrate a 3D array ($\mathbf{X}$) with the shape of (3, 2, 4). Each cell presents an element of the array. Inside each cell is the index of that element, which is denoted by square-bracket tuples, i.e., $\mathbf{X}$ [$i, j, k$]. All the indexing conventions of *NumPy* [20] are adopted here, so the first element is indexed by 0. And the indexing $\mathbf{X}$[1:,0,1::2] represents the highlighted sliced elements in Fig 2. Array elements are stored in computer memory contiguously, and a row-major order is

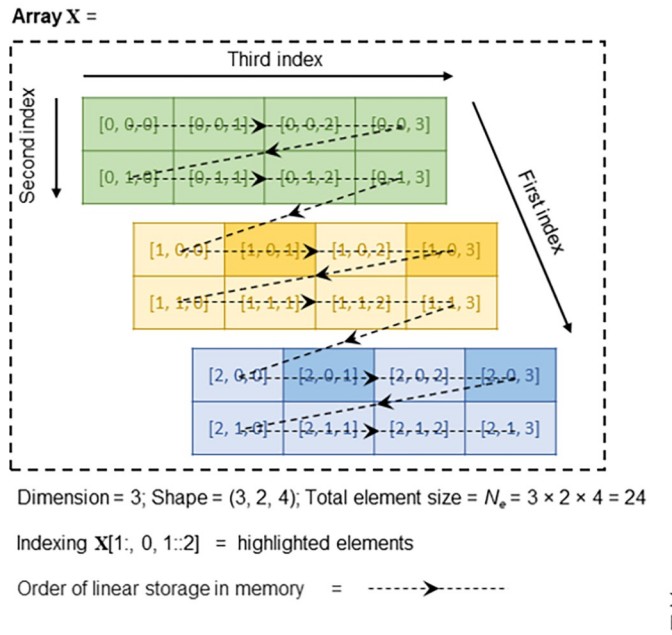

**Fig 2. Array, neighbourhood and representation of neighbouring elements.**

assumed in this study–consecutive elements in the last index are contiguous in memory (indicated by the arrows in Fig 2).

## 2.1 Scope of examined numerical models

This study focuses on numerical models on structured meshes or regular grids. Hence, after discretisation, all the variables (either solution or auxiliary) can be conveniently expressed as arrays–$N$-dimensional arrays for variables in $N$-dimensional problems. Arrays and variables are thus used interchangeably in further discussions. This already represents a broad category of numerical models that can solve many problems [2, 8, 21], that are expressed in linear and non-linear differential equations. Some examples are the finite-difference models on regular grids [22], finite-volume/finite-element models on structured meshes [2, 8], lattice Boltzmann methods [23], etc.

Most numerical models are formatted in the style that in each step, the solution variables $\mathbf{X}_1^{t+1}, \mathbf{X}_2^{t+1}, \ldots$ at next "timestep" ($t + 1$) are updated from the variables $\mathbf{X}_1^t, \mathbf{X}_2^t, \ldots$ at "timestep" $t$. The "timesteps" not only mean the marching of solution variables in time (like most explicit models do) but also can represent the update of solution variables in iterative methods. For complicated models, auxiliary variables must be introduced, and their corresponding arrays allocated in computer memory. Therefore, a substep of a numerical model is defined as a function like Eq 1 such that its implementation is possible without introducing new variables/arrays except for the input and output variables/arrays.

$$\mathbf{Y}_1, \mathbf{Y}_2, \ldots, \mathbf{Y}_{N_0} = F(\mathbf{X}_1, \mathbf{X}_2, \ldots, \mathbf{X}_{N_i}, \boldsymbol{p}) \qquad \text{Eq 1}$$

Here, $N_o$ is the number of output arrays, $N_i$ is the number of input arrays, and $\boldsymbol{p}$ is to indicate all model parameters (non-arrays). The total number of elements for each input/output array depends on the mesh size and is identical for all input and output arrays of substeps examined in this study (denoted as $N_e$).

## 2.2 Maximally fused substeps

Each model step is often fulfilled by several or many substeps. Substeps can have different complexities, and complex ones can always be broken into simpler ones until a handful of very basic ones are obtained like the *NumPy* basic array operations [20]. However, the performance of many simple substeps is always poorer than a fused substep due to the more memory operations involved (demonstrated in Section 5.3). So, to have computationally efficient numerical models, we want to have maximally fused substeps, for which no further optimisation of fusion is possible, and each model step is fulfilled by only a few of these maximally fused substeps.

The detailed specification of Eq 1 is often called a numerical scheme. It depends on the differential equations and discretisation techniques. In general, it takes the format that the elements of output arrays are only locally related to neighbouring elements of input arrays in a fixed pattern (the stencil). Common stencils are the von Neumann neighbourhood and Moore neighbourhood (Fig 2). If we denote $\mathbf{X}[i, j, k]^{<r}$ as all the elements in the array $\mathbf{X}$ that have a Manhatten distance smaller than $r$ regarding the element $\mathbf{X}[i, j, k]$, then a substep is often specified by an equation like Eq 2.

$$\mathbf{Y}_q[i, \ldots] = f_q(\mathbf{X}_1[i, \ldots]^{<r}, \mathbf{X}_2[i, \ldots]^{<r}, \ldots, \mathbf{X}_{N_i}[i, \ldots]^{<r}, \boldsymbol{p}) \ q = 1, 2, \ldots, N_o \qquad \text{Eq 2}$$

Eq 2 only defines the inner elements of $\mathbf{Y}_q$. For a specific numerical model, boundary conditions are required to define the boundary elements.

**Table 1. The examined operations.**

| Label | Input arrays | Output arrays | Parameters (non-arrays) | Numerical scheme equations | JAX implementation | FLOP coef. $\alpha$ | Memory operation coef. $\beta_{lo}$ (= $N_i$+$N_o$) | Memory operation coef. $\beta_{hi}$ (= $\beta$—$\beta_{lo}$) | Required memory coef. $\gamma$ |
|---|---|---|---|---|---|---|---|---|---|
| COPY1D | $\mathbf{X}$ | $\mathbf{Y}$ | N/A | $\mathbf{Y}[i] = \mathbf{X}[i]$ | y = jnp.copy(x) | 0 | 2 (= 1+1) | 0 | 2 |
| COPY2D | | | | $\mathbf{Y}[i,j] = \mathbf{X}[i,j]$ | | | | | |
| SCALE1D | $\mathbf{Y}^t$ | $\mathbf{Y}^{t+1}$ | a | $\mathbf{Y}^{t+1}[i] = a\mathbf{Y}^t[i]$ | y = a * y | 1 | 2 (= 1+1) | 0 | 1 |
| SCALE2D | | | | $\mathbf{Y}^{t+1}[i,j] = a\mathbf{Y}^t[i,j]$ | | | | | |
| AXPY1D | $\mathbf{X}, \mathbf{Y}^t$ | $\mathbf{Y}^{t+1}$ | a | $\mathbf{Y}^{t+1}[i] = \mathbf{Y}^t[i] +a\mathbf{X}[i]$ | y = a * x + y | 2 | 3 (= 2+1) | 0 | 2 |
| AXPY2D | | | | $\mathbf{Y}^{t+1}[i,j] = \mathbf{Y}^t[i,j] +a\mathbf{X}[i,j]$ | | | | | |
| XPXPYN_1D | $\mathbf{X}, \mathbf{Y}^t$ | $\mathbf{Y}^{t+1}$ | N/A | Eq 3 | y = x–x + ... + x—x + y | N | 3 (= 2+1) | 0 | 2 |
| XPXPYN_2D | | | | | | | | | |
| HEAT1D | $\mathbf{T}^t$ | $\mathbf{T}^{t+1}$ | r | Eq 4 | Three methods (see Table 2) | 6 | 2 (= 1+1) | 2 | 2 |
| HEAT2D | | | | Eq 5 | | 8 | 2 (= 1+1) | 4 | 2 |
| NS2D_P | $\mathbf{U}^t, \mathbf{V}^t, \mathbf{P}^t$ | $\mathbf{U}^{t^*}, \mathbf{V}^{t^*}, \mathbf{P}^{t^*}$ | $\Delta t/\Delta x, \Delta t/\Delta y, v\Delta t/(\Delta x)^2, v\Delta t/(\Delta y)^2, c^2\Delta t/\Delta x, c^2\Delta t/\Delta y$ | Eq 7 | Nonlinear differential equations, so convolution is not possible | 46 | 6 (= 3+3) | 10 | 6 |
| NS2D_C | $\mathbf{U}^t, \mathbf{V}^t, \mathbf{P}^t, \mathbf{U}^{t^*}, \mathbf{V}^{t^*}, \mathbf{P}^{t^*}$ | $\mathbf{U}^{t+1}, \mathbf{V}^{t+1}, \mathbf{P}^{t+1}$ | | Eq 8 | | 52 | 9 (= 6+3) | 10 | 6 |
| NS2D | N/A | | | | | 98 | 15 | 20 | 6 |

Eq 1 and Eq 2 are the general representation of model substeps and numerical schemes. The implementation of the substeps as computer programs is interchangeably called functions/operations/kernels/ops. In this paper, operations are simply used. Some examined operations in this study are explained in the next subsections, and a summary of them is given in Table 1.

## 2.3 Element-wise operations

If the output array elements are only related to input array elements at the same place (i.e., the Manhatten distance of neighbouring = 0), these kinds of operations are element-wise operations. Some basic ones are COPY, SCALE, and AXPY (Table 1). Both the vector version and matrix version of these operations are examined. To examine how the number of float-point calculations affects the computing speed, an element-wise operation as Eq 3 is examined and is labelled as XPXPYN, where N is a variable integer number, and it controls the number of float-point calculations.

$$\mathbf{Y}^{t+1}[i, \ldots] = \overbrace{\mathbf{X}[i, \ldots] - \mathbf{X}[i, \ldots]}^{repeat\frac{N}{2}times} + \mathbf{Y}^t[i, \ldots] \qquad \text{Eq 3}$$

Array programming of these operations is straightforward (JAX implementation in Table 1).

## 2.4 Finite-difference model to the 1D heat equation (HEAT1D)

The heat equation is a parabolic partial differential equation modelling the variation of temperature under thermal conduction. The following numerical scheme with only one substep (Eq 4) is obtained if a 1D problem (from 0 to $L$) is discretised into $N_e$ -1 equally spaced segments, the derivative in time is approximated with a forward finite difference, and the second

derivative in space is approximated with a central finite difference.

$$\mathbf{T}^{t+1}[i] = (1 - 2r)\mathbf{T}^t[i] + r(\mathbf{T}^t[i-1] + \mathbf{T}^t[i+1]) \qquad \text{Eq 4}$$

The discretised solution on the nodes is a 1D array $\mathbf{T}^t$ of size $N_e$. The parameter $r = a\Delta t/(\Delta x)^2$, where $a$ is the thermal diffusivity, $\Delta t$ is the time step size, and $\Delta x = L/(N_s\text{-}1)$ is the segment size. This explicit scheme is stable only if $r < 0.5$. Fig 3A shows the solution to Eq 4 with an initial condition of $\mathbf{T}^0[i] = 6\sin(\pi i\Delta x)$, a Dirichlet boundary condition of $\mathbf{T}^{t+1}[0] = 0$ and $\mathbf{T}^{t+1}[-1] = 0$, and parameters of $a = 1.0$, $r = 0.4$, $L = 1.0$, $N_e = 512$. The analytical solution to this problem is $\mathbf{T}^t[i] = 6\sin(\pi i\Delta x)e^{-a\pi\pi t}$.

Three implementations of the 1D heat model with array programming are examined (Table 2): (1) slice and concatenation. The inner elements of $\mathbf{T}^{t+1}$ (of size $N_e$—2) are defined by Eq 4, we can use the slice operations to select three sub-arrays of $\mathbf{T}^t$ (of size $N_e$— 2) first and use basic vector arithmetic operations to calculate the inner elements. Then concatenation is used to combine the inner elements with boundary values to have the final $\mathbf{T}^{t+1}$. (2) Convolution. The heat equation is a linear differential equation, so the inner elements of $\mathbf{T}^{t+1}$ can be obtained by applying a convolution operation to $\mathbf{T}^t$ with a filter of $[r, (1-2.0 \times r), r]$.

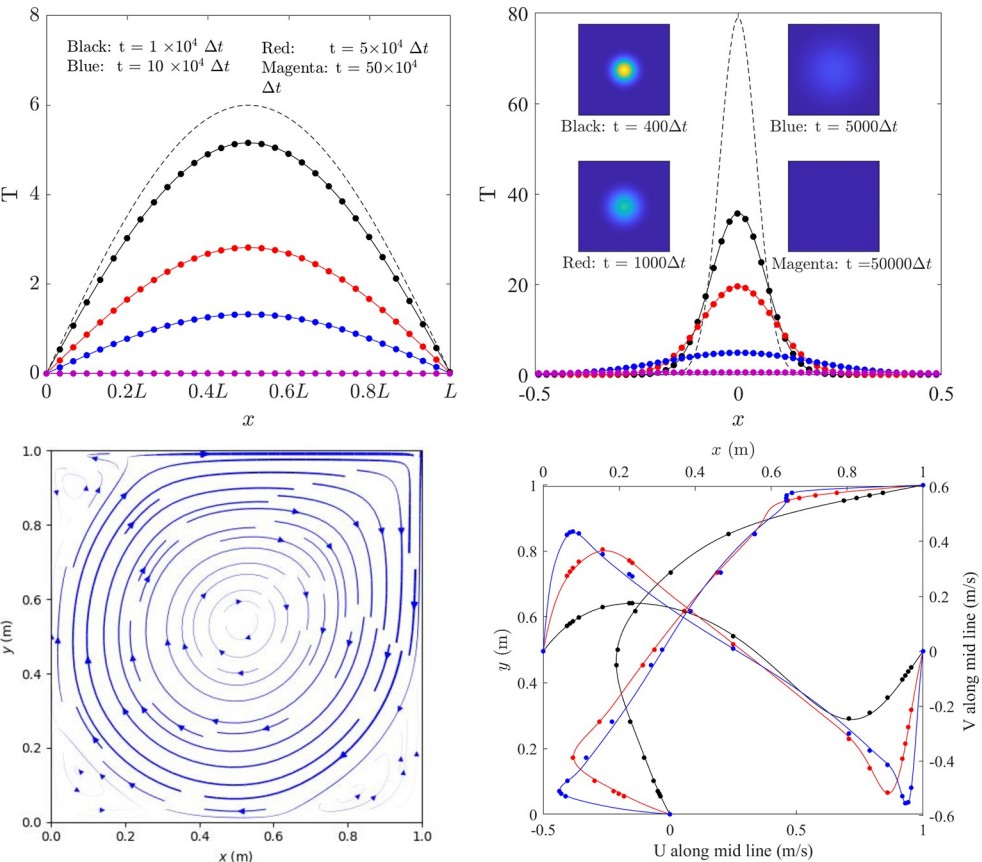

**Fig 3. Boundary-value problems and solutions for the examined numerical models.** (a) Solution to the 1D heat equation with Dirichlet boundary conditions (the dashed line = initial condition; solid lines = analytical solutions; dots = numerical solutions). **(b)** Solution along the midline to the 2D heat equation with Dirichlet boundary conditions (dashed line = initial condition; solid lines = analytical solutions, dots = numerical solutions; insets = temperature contours). (c) Calculated streamline of cavity flow with $Re$ = 5000 (thickness indicates velocity magnitude). (d) Velocity along the midlines (**U** vs. $y$ and **V** vs. $x$; solid lines = numerical solution from this study, dots = numerical solution from Ghia *et al.* [21]; black $Re$ = 100; red $Re$ = 1000; blue $Re$ = 5000).

**Table 2. Implementation of the 1D heat model with array programming.**

| Setup of arrays | | zero = jnp.array([0.0], dtype = #the chosen floating-point type#)<br>r = jnp.array([0.4], dtype = #the chosen floating-point type #)<br>x = jnp.linspace(0.0, 1.0, #No. of nodes#, #the chosen floating-point type #))<br>T = 6.0 * jnp.sin(math.pi * x) |
|---|---|---|
| Different implementations | Slice and concatenation | def substep(T, r, zero):<br>inner = (1–2.0 * r) * T[1:-1] + r * (T[:-2] + T[2:])<br>return jnp.concatenate([zero, inner, zero], axis = 0) |
| | Convolution | filter = jnp.array([r, (1–2.0 * r), r]).squeeze()<br>def substep(T, filter, zero):<br>inner = jnp.convolve(u,filter,mode = 'VALID')<br>return jnp.concatenate([zero, inner, zero], axis = 0) |
| | Roll | def substep(T, r):<br>Tn = (1–2.0 * r) * T + r * (jnp.roll(T, [–1], axis = 0) + jnp.roll(T, [1], axis = 0))<br>Tn = Tn.at[0].set(0.0)<br>Tn = Tn.at[–1].set(0.0)<br>return Tn |

Concatenation is still required to have the final $\mathbf{T}^{t+1}$. (3) Roll. We can ignore the boundary conditions first and use the roll operation to obtain arrays of $\mathbf{T}^t$ (of size $N_e$) that are shifted by one position to the left or to the right. Basic vector arithmetic operations are then used to calculate $\mathbf{T}^{t+1}$, and boundary values are corrected at the end.

## 2.5 Finite-difference model to the 2D heat equation (HEAT2D)

If a 2D square domain (side length = $L$) is discretised into a regular grid with a mesh size $\Delta x = \Delta y = L/(N_x\text{-}1)$, and the same finite-difference approximations are used for the 2D heat equation, the following numerical scheme (Eq 5) is obtained with only one substep:

$$\mathbf{T}^{t+1}[i,j] = (1 - 4r)\mathbf{T}^t[i,j] + r(\mathbf{T}^t[i-1,j] + \mathbf{T}^t[i+1,j] + \mathbf{T}^t[i,j+1] + \mathbf{T}^t[i,j+1]) \qquad \text{Eq 5}$$

The discretised solution on nodes is a 2D array $T^t$ with the number of elements as $N_e = N_x \times N_x$. Similarly, $r = a\Delta t/(\Delta x)^2$ and $\Delta x = L/(N_x\text{-}1)$. Fig 3B shows the solution to Eq 5 within a square domain (from -1.0 to 1.0 in both directions and $L = 2.0$) subjected to the initial condition of Eq 6 and zero-temperature boundary conditions. Parameters are $a = 1.0$, $r = 0.2$, $N_x = 512$ and $t_0 = 0.001$. The analytical solution to this problem is also shown in Eq 6.

$$\mathbf{T}^0(x,y) = \frac{e^{-\frac{x^2+y^2}{4at_0}}}{4\pi at_0}$$

$$\qquad \text{Eq 6}$$

$$\mathbf{T}(x,y,t) = \frac{e^{-\frac{x^2+y^2}{4a(t+t_0)}}}{4\pi a(t+t_0)}$$

With similar techniques as in Table 2, three methods are available to implement this 2D model with array programming, the detail is omitted here.

## 2.6 MacCormack scheme to 2D Navier-Stokes equation with artificial compressibility (NS2D)

If a 2D rectangular domain (with side lengths $L_x$ and $L_y$) is discretised into a regular grid with mesh sizes $\Delta x = L_x/(N_x\text{-}1)$ and $\Delta y = L_y/(N_y\text{-}1)$. The solution variables/arrays to the 2D Navier-Stokes equation are the horizontal component of velocity ($\mathbf{U}^t$), the vertical component of

velocity ($\mathbf{V}^t$) and the pressure ($\mathbf{P}^t$)–all with the shape of ($N_x$, $N_y$). The total number of elements for each array is $N_e = N_x \times N_y$. If the artificial compressibility method is used and the equations are discretised using the MacCormack scheme, the model is made of two maximally fused sub-steps [24].

For the first substep (often called the predictor substep), the input arrays are $\mathbf{U}^t$, $\mathbf{V}^t$, $\mathbf{P}^t$ and the output arrays are "provisional" variables/arrays $\mathbf{U}^{t^*}$, $\mathbf{V}^{t^*}$, $\mathbf{P}^{t^*}$. The detailed numerical scheme is in Eq 7, in which all the first-order derivatives in space are approximated with a forward finite difference and second-order derivatives are approximated with a central finite difference.

$$\mathbf{U}^{t*}[i,j] = \mathbf{U}^t[i,j]$$
$$-\frac{\Delta t}{\Delta x}(\mathbf{U}^t[i,j](\mathbf{U}^t[i+1,j] - \mathbf{U}^t[i,j]) + \mathbf{P}^t[i+1,j] - \mathbf{P}^t[i,j])$$
$$-\frac{\Delta t}{\Delta y}\mathbf{V}^t[i,j](\mathbf{U}^t[i,j+1] - \mathbf{U}^t[i,j])$$
$$+v\frac{\Delta t}{(\Delta x)^2}(\mathbf{U}^t[i+1,j] - 2\mathbf{U}^t[i,j] + \mathbf{U}^t[i-1,j])$$
$$+v\frac{\Delta t}{(\Delta y)^2}(\mathbf{U}^t[i,j+1] - 2\mathbf{U}^t[i,j] + \mathbf{U}^t[i,j-1])$$

Eq 7A

$$\mathbf{V}^{t*}[i,j] = \mathbf{V}^t[i,j]$$
$$-\frac{\Delta t}{\Delta x}(\mathbf{U}^t[i,j](\mathbf{V}^t[i+1,j] - \mathbf{V}^t[i,j]))$$
$$-\frac{\Delta t}{\Delta y}(\mathbf{V}^t[i,j](\mathbf{V}^t[i,j+1] - \mathbf{V}^t[i,j]) + \mathbf{P}^t[i,j+1] - \mathbf{P}^t[i,j])$$
$$+v\frac{\Delta t}{(\Delta x)^2}(\mathbf{V}^t[i+1,j] - 2\mathbf{V}^t[i,j] + \mathbf{V}^t[i-1,j])$$
$$+v\frac{\Delta t}{(\Delta y)^2}(\mathbf{V}^t[i,j+1] - 2\mathbf{V}^t[i,j] + \mathbf{V}^t[i,j-1])$$

Eq 7B

$$\mathbf{P}^{t*}[i,j] = \mathbf{P}^t[i,j]$$
$$-c^2\frac{\Delta t}{\Delta x}(\mathbf{U}^t[i+1,j] - \mathbf{U}^t[i,j])$$
$$-c^2\frac{\Delta t}{\Delta y}(\mathbf{V}^t[i,j+1] - \mathbf{V}^t[i,j])$$

Eq 7C

In Eq 7, $v$ is the fluid dynamic viscosity, $c$ is an artificial constant representing the speed of sound, and $\Delta t$ is the timestep, which must meet the stability criteria ($\Delta t < C_{max}\Delta x/c$ and $\Delta t < C_{max}(\Delta x)^2/v$).

For the second substep (often called the corrector substep), the input arrays are solution arrays from the previous timestep $\mathbf{U}^t$, $\mathbf{V}^t$, $\mathbf{P}^t$ and the "provisional" arrays $\mathbf{U}^{t^*}$, $\mathbf{V}^{t^*}$, $\mathbf{P}^{t^*}$. The output arrays are solution arrays for the next timestep $\mathbf{U}^{t+1}$, $\mathbf{V}^{t+1}$, $\mathbf{P}^{t+1}$. In the corrector substep, the first-order derivatives in space are approximated with the backward finite difference, and

the detailed numerical scheme is in Eq 8.

$$\mathbf{U}^{t+1}[i,j] = \frac{1}{2}\{\mathbf{U}^t[i,j] + \mathbf{U}^{t*}[i,j]$$

$$-\frac{\Delta t}{\Delta x}(\mathbf{U}^{t*}[i,j](\mathbf{U}^{t*}[i,j] - \mathbf{U}^{t*}[i-1,j]) + \mathbf{P}^{t*}[i,j] - \mathbf{P}^{t*}[i-1,j])$$

$$-\frac{\Delta t}{\Delta y}\mathbf{V}^{t*}[i,j](\mathbf{U}^{t*}[i,j] - \mathbf{U}^{t*}[i,j-1])$$

$$+v\frac{\Delta t}{(\Delta x)^2}(\mathbf{U}^{t*}[i+1,j] - 2\mathbf{U}^{t*}[i,j] + \mathbf{U}^{t*}[i-1,j])$$

$$+v\frac{\Delta t}{(\Delta y)^2}(\mathbf{U}^{t*}[i,j+1] - 2\mathbf{U}^{t*}[i,j] + \mathbf{U}^{t*}[i,j-1])\}$$

Eq 8A

$$\mathbf{V}^{t+1}[i,j] = \frac{1}{2}\{\mathbf{V}^t[i,j] + \mathbf{V}^{t*}[i,j]$$

$$-\frac{\Delta t}{\Delta x}(\mathbf{U}^{t*}[i,j](\mathbf{V}^{t*}[i,j] - \mathbf{V}^{t*}[i-1,j]))$$

$$-\frac{\Delta t}{\Delta y}(\mathbf{V}^{t*}[i,j](\mathbf{V}^{t*}[i,j] - \mathbf{V}^{t*}[i,j-1]) + \mathbf{P}^{t*}[i,j] - \mathbf{P}^{t*}[i,j-1])$$

$$+v\frac{\Delta t}{(\Delta x)^2}(\mathbf{V}^{t*}[i+1,j] - 2\mathbf{V}^{t*}[i,j] + \mathbf{V}^{t*}[i-1,j])$$

$$+v\frac{\Delta t}{(\Delta y)^2}(\mathbf{V}^{t*}[i,j+1] - 2\mathbf{V}^{t*}[i,j] + \mathbf{V}^{t*}[i,j-1])\}$$

Eq 8B

$$\mathbf{P}^{t+1}[i,j] = \frac{1}{2}\{\mathbf{P}^t[i,j] + \mathbf{P}^{t*}[i,j]$$

$$-c^2\frac{\Delta t}{\Delta x}(\mathbf{U}^{t*}[i,j] - \mathbf{U}^{t*}[i-1,j])$$

$$-c^2\frac{\Delta t}{\Delta y}(\mathbf{V}^{t*}[i,j] - \mathbf{V}^{t*}[i,j-1])\}$$

Eq 8C

A particular application of this numerical scheme is the cavity flow problem (Fig 3C and 3D). A square domain (side length = 1 m) is filled with fluids, which is driven by the top lid. So, the top boundary has constant velocity (U = $u_0$ = 1 m/s and V = 0), and the other boundaries have zero velocities. Neumann boundary condition is used for the pressure field (i.e., $\partial P/\partial n = 0 \rightarrow P^{t+1}[0,:] = P^{t+1}[1,:]$, ...). Simulations start with homogenous variables (U = 0, V = 0, and P = 0), and the final steady-state velocity field depends on the Reynolds number Re = $u_0 L_x/v$ [24]. Fig 3C shows the streamlines for $Re$ = 5000, which clearly shows a primary vertex and three small vortices at corners. Fig 3D shows the velocities along the mid lines from this study and results from Ghia $et$ $al.$ [21]. The speed of sound is chosen as $c = 0.1u_0$ and a steady state is achieved after $2 \times 10^5$, $7 \times 10^5$, and $35 \times 10^5$ increments for $Re$ = 100, 1000, and 5000, respectively. Similarly, this model can be implemented with array programming. However, because this model is non-linear, the method of using convolution operations is not possible.

**Table 3. Information of the tested CPU platforms.**

| | General information | | FLOPS | | | | | Bandwidth |
|---|---|---|---|---|---|---|---|---|
| | | Instructions | | 128-bit SSE | | 256-bit AVX | | Theoretical "burst" rate (GB/s) |
| | | Floating-point number | | f32 | f64 | f32 | f64 | |
| | | FLOPs per cycle | | 8 | 4 | 16 | 8 | |
| PC | **CPU:** Name (Intel Core i7-9850H), Sockets (1), Cores per socket (6), Frequency (2.6 GHz), L1 data cache (192 kB), L2 cache (1.5 MB), L3 Cache (12 MB) | Theoretical FLOPS (GFLOPS) | | 124.8 | 62.4 | 249.6 | 124.8 | 88.3 |
| | **Memory:** Frequency (2.67GHz), No. of channels (2), Bus width (64 bits), Total size (2 * 16 GB) | Benchmarked FLOPS (GFLOPS) | | 185.5 | 92.2 | 281.7 | 140.3 | |
| HPC | **CPU**: Name (Intel Xeon Gold 6238R), Sockets (26), Cores per socket (1), Frequency (2.2 GHz), L1 data cache (32 kB), L2 cache (1 MB), L3 Cache (38.5 MB) | Theoretical FLOPS (GFLOPS) | | 457.6 | 228.8 | 915.2 | 457.6 | 256.0 |
| | **Memory:** Frequency (2.67GHz), No. of channels (6), Bus width (64 bits), Total size (88 GB) | Benchmarked FLOPS (GFLOPS) | | 548.2 | 272.5 | 956.9 | 487.3 | |

## 3 Computer performance

In this paper, the optimal implementations of the substeps/operations on both the CPU platform and GPU platform are studied. The performance of such optimal implementations is then examined, modelled, and compared with the performance of XLA implementations. These are conducted on two computers–one personal computer (PC) and one high-performance computing (HPC) workstation. Computation is possible on both the CPU and GPU platforms for these two computers. The PC has an *Intel Core i7-9850H* CPU and an *Nvidia Quadro P620* GPU running on *Ubuntu 20.04.5 LTS*. The HPC has an *Intel Xeon Gold 6238R* CPU and *an Nvidia Quadro RTX 5000* GPU running on *Red Hat Enterprise Linux Workstation 7.9*. The technical specification of the two computers is given in Table 3 (CPU platforms) and Table 4 (GPU platforms).

The execution of the operations costs computer resources, and the execution time depends on computer performance. The two relevant measurements of computer performance for numerical models are floating point operations per second (FLOPS) and bandwidth. The two most common floating-point numbers are the single-precision floating-point numbers (each number occupies 32 bits or 4 bytes) and the double-precision floating-point numbers (64 bits or 8 bytes). They are conveniently denoted as f32 and f64 in the latter discussions.

**Table 4. Information of the tested GPU platforms.**

| | General information | FLOPS | | Bandwidth |
|---|---|---|---|---|
| | | Single precision (GFLOPS) | Double precision (GFLOPS) | Theoretical "burst" rate (GB/s) |
| PC | **GPU:** Name (Nvidia Quadro P620), CUDA capability (6.1), Multiprocessors (4), CUDA Cores per multiprocessors (128), Frequency (1.266 GHz), L2 Cache (512 kB), Shared memory per multiprocessor (96 kB) | 1490 | 46.6 | 96.1 |
| | **GPU Memory:** Frequency (1.003 GHz), No. of channels (2), Bus width (128 bits), Total size (4 GB) | | | |
| HPC | **GPU:** Name (Nvidia Quadro RTX 5000), CUDA capability (7.5), Multiprocessors (48), CUDA Cores per multiprocessors (64), Frequency (1.62 GHz), L2 Cache (4 MB), Shared memory per multiprocessor (64 kB) | 11151 | 348.5 | 448.0 |
| | **GPU Memory:** Frequency (1.75 GHz), No. of channels (4), Bus width (256 bits), Total size (16 GB) | | | |

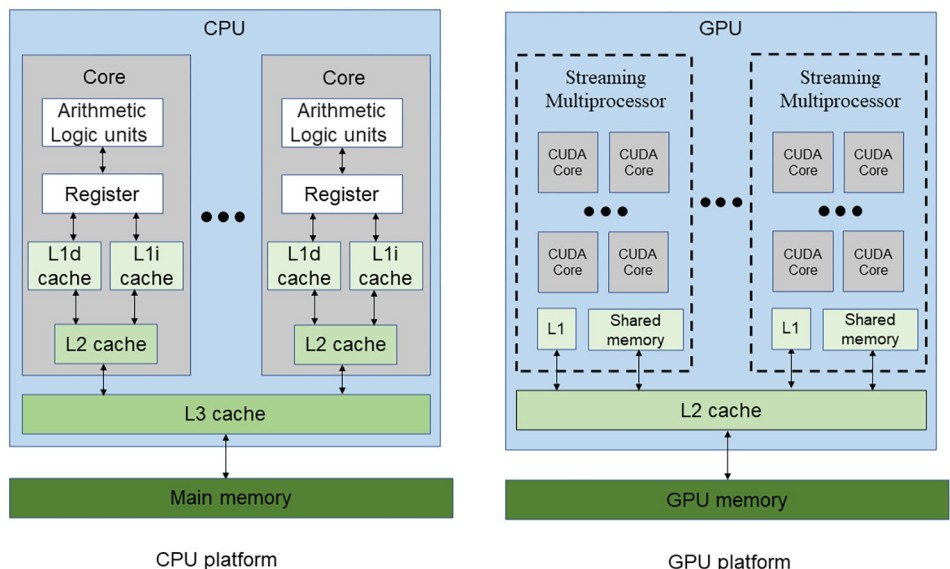

**Fig 4. Architecture of the CPU and GPU computing platforms.**

## 3.1 Floating point operations per second (FLOPS)

As the name indicates, the FLOPS measures the number of floating-point operations (FLOPs) a processor can execute within a second. A typical unit is GFLOPS (i.e., gigaFLOPS and $10^9$ FLOPS).

On the CPU platform, it can be calculated as FLOPS = No. of sockets × No. of cores per socket × CPU frequency × No. of FLOPs per cycle [25]. The number of FLOPs per cycle depends on instruction sets. For example, the 128-bit SSE (streaming SIMD extensions) is one of the SIMD (single instruction multiple data) instruction sets, it can execute 8 FLOPs per cycle for single precision and 4 FLOPs per cycle in double precision (Table 3). The 256-bit AVX (advanced vector extensions) doubles the FLOPs per cycle (Table 3). High FLOPS of the CPUs are achieved by these SIMD instruction sets, and the corresponding values are calculated from the equation and shown in Table 3. It is shown that the FLOPS of 256-bit AVX is twice that of 128-bit SSE, and the FLOPS of single precision calculation is twice that of double precision. The FLOPS can also be benchmarked by computer programs. Table 3 includes the benchmarked FLOPS by an open-source program called *Flops* [26], which is slightly higher than the theoretical values.

The FLOPS of the GPU platforms is provided by *Nvidia* [27] and is listed in Table 4. For the two GPUs, the FLOPS of single precision calculation is significantly higher than that of double precision.

In summary, the FLOPS for each processor can be written as a function of two arguments–FLOPS($s_f$, $x_{is}$), where $s_f$ is the size of a floating-point number (i.e., 4 bytes for f32 and 8 bytes for f64) and $x_{is}$ denotes the instruction sets.

## 3.2 Bandwidth (BW)

The typical architecture of the CPU and GPU computing platforms is illustrated in Fig 4. On both platforms, the calculations (i.e., FLOPs) are performed by the arithmetic logic unit. The data to be operated by the arithmetic logic unit (called operands) reside on the register and so do the operation results. However, the size of the register is very small, most data are stored in

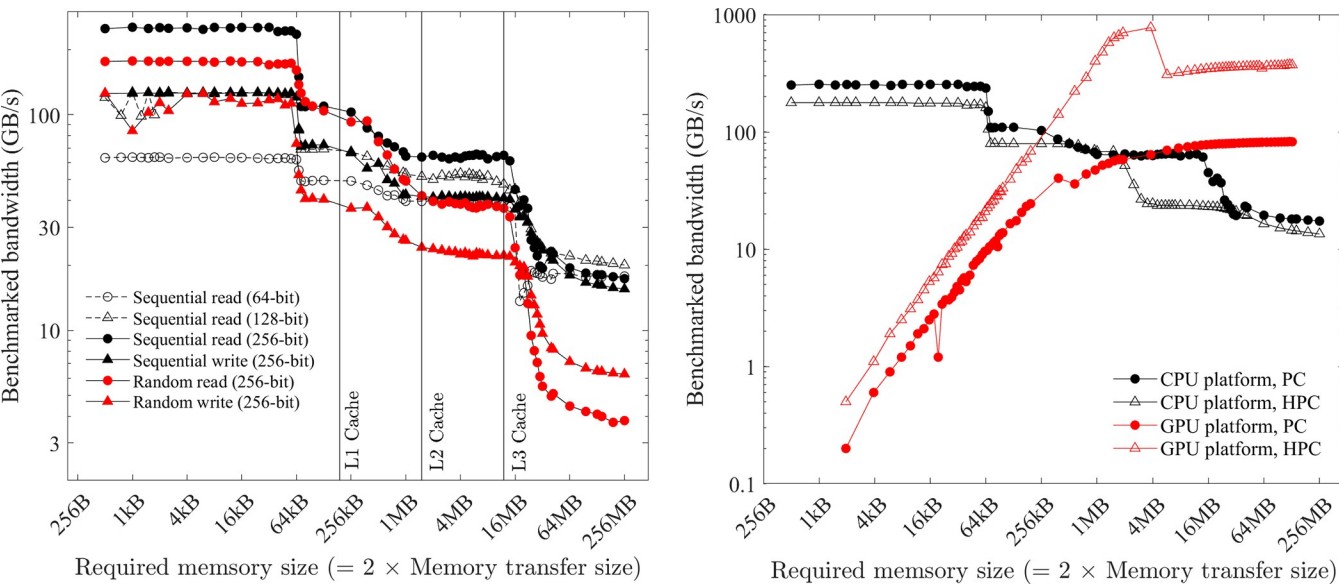

**Fig 5. Benchmarked bandwidth.** (a) from *bandwidth-benchmark* on the PC. (b) CPU platform from *bandwidth-benchmark* for sequential 256-bit read and GPU platform from *bandwidthTest* of CUDA samples with "shmoo" mode.

the memory or cache. So, to finish FLOPs, operands must be read to the register, and the operation results written back to the memory or cache after calculation. Bandwidth is a measure of the rate how data is read or written. A typical unit is GB/s (i.e., gigabytes per second). For many applications, the processor (CPU or GPU) tends to access the same set of memory locations repetitively over a short period (called the locality of reference), so both computing platforms are optimised with a hierarchical memory system (Fig 4)–from L1 cache, L2 cache, L3 cache to the memory with an increasing storage size but decreasing bandwidth.

The theoretical bandwidth of the memory can be calculated as BW = memory frequency × No. of data transfers per cycle × No. of channels × bus width [28]. Here the number of data transfers per cycle is two for double data rate memory (i.e., DDR, DDR2…). This theoretical bandwidth is often referred to as "burst rate" (calculated and listed in Tables 3 and 4) because it is not sustainable. It is more realistic to benchmark the bandwidth with computer programs. Fig 5A gives some results from the open-source software *bandwidth-benchmark* [29] for the CPU platform of the PC. The core routines of this software are written with the low-level assembly language, so the benchmarked result is a good measure of the hardware performance and does not depend on the compiler version or options. The following five observations are typical for CPU platforms:

1. The bandwidth depends on how much data is read/written for each instruction. Sequentially read in 256 bits (i.e., 256-bit AVX; black filled circles in Fig 5A) is faster than read in 128 bits (i.e., 128-bit SSE; black hollow triangles), and the slowest is read in 64 bits (black hollow circles).

2. The bandwidth of reading is slightly faster than writing but the gap is small (black filled circles vs. black filled triangles in Fig 5A).

3. The bandwidth depends on how much total memory is required. When the required memory size is smaller than the L1 cache size, the source and/or the destination can reside on the L1 cache, so the bandwidth is the fastest (~250 GB/s for reading in 256-bit AVX in Fig 5A). With an increased size of required memory, the bandwidth decreases and Fig 5A

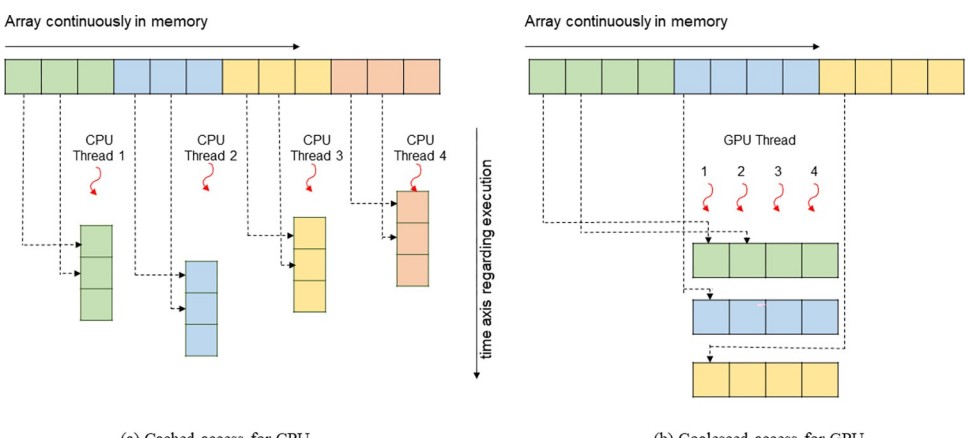

Fig 6. **Memory access pattern and bandwidth.** (a) Cached access for CPU. (b) Coalesced access for GPU.

clearly shows the abrupt drop of bandwidth at three critical positions (i.e., L1, L2 and L3 cache sizes). When the required memory size is larger than the L3 cache size, the copy operation cannot resort to the caches, so the benchmarked bandwidth is the sustained bandwidth of the main memory (about 20 GB/s), which is significantly smaller than its "burst" rate (88 GB/s). Moreover, the bandwidth of the L1 cache is more than 10 times faster than that of the main memory (i.e., 250 GB/s vs. 20 GB/s). Because of this dependence on required memory size, in the latter discussions, a numerical model is roughly classified to be small if the required memory size is smaller than the L1 cache size, medium if between the L1 and L3 cache sizes, and large if larger than the L3 cache size.

4. The bandwidth depends on the memory access pattern. CPU threads are independent and may execute at their own pace. So, the CPU threads prefer cached access (Fig 6A)–if a thread's current access is at a specific position, its preferred next access should be at the sequential next position in memory. From the comparison of the black lines and red lines in Fig 5A, it is seen that the bandwidth of sequential read/write (cached access) is significantly faster than that of random read/write (uncached access).

5. The bandwidth depends on the number of threads used. The bandwidth of reading in 256-bit AVX for both the PC and HPC is shown in Fig 5B. Because the benchmark tool uses only one thread, the measured bandwidth of the HPC is even slower than that of the PC. It is shown in Section 5.1 that using multiple threads can increase the bandwidth.

The bandwidth of the GPU platform has similar five observations:

1. The bandwidth of the GPU platform also depends on how much data is read/written for each instruction. It is shown in Section 5.2 that using the CUDA built-in type *float4* and *double2* (i.e., read/write four f32 or two f64 numbers with one instruction) can gain minor improvement compared with using f32 or f64 (i.e., read/write one f32 or f64 number with one instruction).

2. There is no evidence of a noticeable difference regarding the bandwidth of reading and writing on the GPU platform, so reading and writing are assumed to have equal bandwidth.

3. The bandwidth of the GPU platform also depends on how much total memory is required. The red lines in Fig 5B show the benchmarked bandwidth by the open-source software

*bandwidthTest* (with option "shmoo") from the CUDA samples [30]. When the required memory size is small ($< 64$ kB), the bandwidth is very small ($< 5$ GB/s). The bandwidth gradually increases with the increase of the required memory size from 64 kB to about 16 MB. There is a peak bandwidth of about 900 GB/s for the HPC (at about 4 MB required memory size). When the required memory size is larger than 16 MB, the bandwidth is constant (about 82.4 and 372.3 GB/s for the PC and HPC, respectively) and is slightly smaller than the "burst" rate (96.1 and 448.0 GB/s, respectively). Similarly, on the GPU platform, a numerical model is roughly classified to be small if the required memory size is smaller than 64 kB, medium if between 64 kB and 16 MB, and large if larger than 16 MB.

4. The bandwidth of the GPU platform also depends on the memory access pattern. CPU threads are independent and may execute at their own pace. In contrast, the GPU threads execute synchronously (i.e., threads in groups must execute instructions together), and all threads in a group (warp) must finish their work before any thread can move on [31]. Therefore, the GPU threads prefer coalesced access (Fig 5B)–if a thread's current access is at a specific position, the next thread's preferred current access should be at the sequential next position in memory. It is shown in Section 5.6 that the bandwidth of coalesced access is significantly faster than that of uncoalesced access.

5. The GPU platform is designed to run multiple threads synchronously, and the number of threads is often a multiple of the warp size (32 for both the PC and HPC). So, the commonly used setting is running with 128, 256, 512, or 1024 threads concurrently. In contrast to the CPU platform, the bandwidth of the GPU platform is not affected by the number of threads (at least for the commonly used settings), as shown in Section 5.

In summary, if the small bandwidth gap between reading and writing is ignored, the bandwidth can be written as a function of four arguments–$BW(s_t, x_{is}, x_{ap}, x_{th})$, where $s_t$ is the total size of required memory, $x_{is}$ indicates the instruction set, $x_{ap}$ indicates the memory access pattern, and, $x_{th}$ indicates the use of parallel threads.

## 4 Modelling the computing speed and optimisation strategies

### 4.1 Computing latency of operations

The total number of FLOPs for one execution of the operations can be roughly estimated. If there are $\alpha^q$ FLOPs for the $q$th equation in Eq 2, then approximately $\alpha^q N_e$ FLOPs are needed to calculate $\mathbf{Y}_q$ by ignoring the special equations for the boundary conditions. Here, $N_e$ is the number of elements in an array. The total number of FLOPs is then $\alpha^1 N_e + \alpha^2 N_e + \ldots + \alpha^{No} N_e = \alpha N_e$. $\alpha$ is a dimensionless constant determined by the numerical scheme (Eq 2). For the operations in this study, the $\alpha$ values are presented in Table 1. The time spent on FLOPs is then approximately $t_{flop} = \alpha N_e / FLOPS(s_f, x_{is})$.

The required memory size is also approximately proportional to the number of elements by ignoring the non-array parameters, i.e., the required memory size is $\gamma N_e s_f$. Here $s_f$ is the size of a floating-point number (i.e., 4 bytes for f32 and 8 bytes for f64) and $\gamma$ is a dimensionless constant. In most applications, operations are implemented as in-place updates to reduce the required memory size. For example, for the NS2D operation, six arrays ($\gamma = 6$) are allocated in memory. In the corrector step, the output arrays $\mathbf{U}^{t+1}$, $\mathbf{V}^{t+1}$, $\mathbf{P}^{t+1}$ and input arrays $\mathbf{U}^t$, $\mathbf{V}^t$, $\mathbf{P}^t$ use the same arrays in computer memory and the outputs are updated in place. Similarly, in-place updates are used for AXPY and XPXPYN, so $\gamma = 2$ for them.

The total number of memory operations can also be estimated. For all the $N_o$ equations like Eq 2, the number of writing operations (i.e., write variables from the register to the memory or

cache) is $N_o$, and the number of reading operations $N_r$ equals the number of non-duplicate array elements in the right-hand of the equations (non-duplicate because an element only needs to be read to the register once). For example, to calculate the "provisional" arrays elements $\mathbf{U}^{t^*}[i,j]$, $\mathbf{V}^{t^*}[i,j]$, $\mathbf{P}^{t^*}[i,j]$ ($N_o = 3$) with Eq 7 in the predictor substep, 13 ($N_r$) reading operations are needed– 5 for $\mathbf{U}^t$ (i.e., $\mathbf{U}^t[i,j]$, $\mathbf{U}^t[i+1,j]$, $\mathbf{U}^t[i-1,j]$, $\mathbf{U}^t[i,j+1]$, $\mathbf{U}^t[i,j-1]$), 5 for $\mathbf{V}^t$, and 3 for $\mathbf{P}^t$ (i.e., $\mathbf{P}^t[i,j]$, $\mathbf{P}^t[i+1,j]$, $\mathbf{P}^t[i,j+1]$,). Therefore, the total number of memory operations is approximately $N_o N_e s_f + N_r N_e s_f = \beta N_e s_f$. For element-wise operations, the number of non-duplicate elements in the right-hand equals the number of input arrays $N_i$, so $\beta = N_i + N_o$. In implementations, it is not optimal to conduct these memory operations at the same bandwidth, so the workload is split into two parts–one part operated at a lower bandwidth ($\beta_{lo} N_e s_f$), the other part at a higher bandwidth ($\beta_{hi} N_e s_f$), and $\beta = \beta_{lo} + \beta_{hi}$. The time spent on memory operations is then approximately $t_m = t_{mlo} + t_{mhi} = \beta_{lo} N_e s_f / \mathrm{BW_{lo}}(s_t = \gamma N_e s_f, x_{is}, x_{ap}, x_{th}) + \beta_{hi} N_e s_f / \mathrm{BW_{hi}}(s_t = \gamma N_e s_f, x_{is}, x_{ap}, x_{th})$.

An operation is called memory-bound if the time spent on memory operations (mostly on low-bandwidth operations) is larger than that on FLOPs (i.e., $t_{mlo} > t_{flop}$). Otherwise, it is FLOP-bound. Therefore, an operation is memory-bound if $\alpha/\beta_{lo} < s_f \mathrm{FLOPS}/\mathrm{BW_{lo}}$. The left-hand side of the inequality is a ratio of workloads between FLOPs and memory operations, which is determined by the numerical scheme. The right-hand side is a hardware parameter, and a new symbol is used for it–$(\alpha/\beta_{lo})_c$. This hardware parameter represents a critical ratio, and an operation is memory-bound if $\alpha/\beta_{lo} < (\alpha/\beta_{lo})_c$. Additionally, the ratio of time spent on FLOPs and low-bandwidth memory operations is $t_{flop}/t_{mlo} = [\alpha/\beta_{lo}]/[(\alpha/\beta_{lo})_c]$.

Similarly, the time spent on memory operations is mostly due to low-bandwidth read/write (i.e., $t_{mlo} > t_{mhi}$) when $\beta_{hi}/\beta_{lo} < \mathrm{BW_{hi}}/\mathrm{BW_{lo}}$. Similarly, the left-hand side is a ratio of workload, the right-hand side is a hardware parameter, and a new symbol is used for it–$(\beta_{hi}/\beta_{lo})_c$. The ratio of time spent on high- and low-bandwidth memory operations is $t_{mhi}/t_{mlo} = [\beta_{hi}/\beta_{lo}]/[(\beta_{hi}/\beta_{lo})_c]$.

In parallel computing, the coordination of all the parallel threads costs computer resources and thus time. Additionally, the read of non-array parameters costs bandwidth. These and many other tasks do not scale with the mesh/grid size ($N_e$), and the time spent on them is collectively denoted as the overhead time $t_{oh}$. Because of this non-scaling, the overhead time is often negligible compared to the time on low-bandwidth memory operations ($t_{oh}/t_{mlo} << 1$) when the mesh size $N_e$ is large.

The computing speed of an operation can be measured in latency (LT; the time in seconds needed for one execution of the operation) and throughput (THP; the number of operations executed within a second). In most scenarios of the present study, the computing time is mostly spent on low-bandwidth memory operations, i.e., $t_{mlo} = \max(t_{flop}, t_{mlo}, t_{mhi}, t_{oh})$, and LT $\geq t_{mlo}$. Additionally, these workloads may run in parallel (e.g., the processor may conduct FLOPs, and at the same time read array elements needed for the next FLOPs), so the LT is smaller than the sum of these estimated times, i.e., LT $\leq t_{flop} + t_{mlo} + t_{mhi} + t_{oh}$. The following equation is therefore obtained.

$$\frac{\beta_{lo} N_e s_f}{\mathrm{BW_{lo}}} = t_{mlo} \leq \mathrm{LT} = \frac{1}{\mathrm{THP}} \leq t_{flop} + t_{mlo} + t_{mhi} + t_{oh}$$

$$= \frac{\alpha N_e}{\mathrm{FLOPS}} + \frac{\beta_{lo} N_e s_f}{\mathrm{BW_{lo}}} + \frac{\beta_{hi} N_e s_f}{\mathrm{BW_{hi}}} + t_{oh}$$

Eq 9

## 4.2 Optimisation strategies

The size $N_e$ is determined by the discretised mesh/grid and is thus fixed. The memory size of a floating-point number $s_f$ (4 bytes for f32 and 8 bytes for f64) is determined by the accuracy

requirements. So, the required memory size of a numerical model is fixed ($\gamma N_e s_f$), and so does its relative size (e.g., small, medium, or large).

From Eq 9, to reduce latency and to increase throughput, there are two categories of approaches: optimise the numerical scheme (the coefficients $\alpha$ and $\beta$ are determined by the numerical scheme) and optimise the implementation on target computing platforms (the FLOPS and BW are hardware parameters that depend on implementation details).

The optimisation of the numerical scheme is similar to the target-independent optimisation in XLA and includes the following strategies: (1) Optimisations to have maximally fused sub-steps, which is explained in Section 2. Fused operations can reduce memory operations and are one of the best ways to improve performance (demonstrated in Section 5.3). (2) Optimisations to reduce the number of FLOPs, i.e., reduce $\alpha$. (3) Optimisations to reduce the number of memory operations, i.e., reduce $\beta$. The numerical models examined in this study are already optimised with maximally fused substeps and Eq 4~Eq 8 are already expressed in the optimal format with the smallest values regarding $\alpha$ and $\beta$. The element-wise operations are also optimal except for XPXPYN. After optimisation, XPXPYN is $\mathbf{Y}^{t+1} = \mathbf{Y}^t$, which has smaller $\alpha$ (0) and $\beta$ (2). However, XPXPYN is designed to examine how the number of FLOPs influences computing speed. So, in this study, the numerical scheme equations are assumed already optimised, and $\alpha$ and $\beta$ are simply used to denote the smallest values of the optimised numerical schemes.

The optimisation of implementation depends on the target computing platform and includes the following strategies: (1) Optimisations to increase FLOPS. It is shown in Section 3.1 that the FLOPS is a function of two arguments–FLOPS($s_f$, $x_{is}$). The size of the floating-point number $s_f$ is determined by the accuracy requirements, so the optimal implementation should choose the appropriate instruction sets to have FLOPS$^{max}$($s_f$) = max{FLOPS($s_f$, $x_{is}$)}. (2) Optimisations to increase bandwidth. Similarly, it is shown in Section 3.2 that the bandwidth is a function of four arguments–BW($s_t = \gamma N_e s_f$, $x_{is}$, $x_{ap}$, $x_{th}$). The total required memory size $s_t = \gamma N_e s_f$ is fixed by the problem. So the optimal implementation should choose the appropriate methods to have BW$_{lo}^{max}$($s_t = \gamma N_e s_f$) = max{BW$_{lo}$($s_t = \gamma N_e s_f$, $x_{is}$, $x_{ap}$, $x_{th}$)}and BW$_{hi}^{max}$($s_t = \gamma N_e s_f$). (3) Optimisations to reduce low-bandwidth memory operations: After the optimisation of the numerical scheme, the total number of memory operations (i.e., $\beta$) is fixed. Higher computing speed is still achieved by reducing memory operations at lower bandwidth (i.e.., reducing $\beta_{lo}$ and increasing $\beta_{hi}$). (4) Optimisations to reduce the overhead time.

After optimisations, the latency is minimised, and the throughput is maximised. They can be estimated by substituting the optimal hardware parameters (e.g., FLOPS$^{max}$, BW$_{lo}^{max}$, and BW$_{hi}^{max}$) in Eq 9.

## 4.3 Effective bandwidth

The input arrays are at least read once at low bandwidth and the output arrays are at least written once, so the minimal possible value for the coefficient $\beta_{lo}$ is at least $N_i + N_o$. It is shown in Section 5 that optimal implementations do have $\beta_{lo} = N_i + N_o$ and $\beta_{hi} = \beta - (N_i + N_o)$. Therefore, for a fixed mesh size ($N_e$) with a fixed accuracy requirement ($s_f$), the minimum workload of low-bandwidth memory operations is fixed at $\beta_{lo} N_e s_f = (N_i + N_o) N_e s_f$. The computing speed can then be measured by another quantity–the effective bandwidth BW$_e$ = $\beta_{lo} N_e s_f$/LT = $(N_i + N_o) N_e s_f$/LT = $(N_i + N_o) N_e s_f \times$THP, which is a measure of throughput in terms of memory operations. From the equation, the latency, throughput, and effective bandwidth are all equivalent and it is easy to calculate one from another. However, the use of effective bandwidth has the advantage that it can be compared with some reference values (e.g., benchmarked bandwidth, theoretical burst rate, etc.).

With these definitions, the following equation regarding the effective bandwidth $BW_e$ is obtained from Eq 9.

$$\zeta BW_{lo} \leq BW_e \leq BW_{lo}$$

$$\zeta = \frac{1}{1 + \dfrac{t_{flop}}{t_{mlo}} + \dfrac{t_{mhi}}{t_{mlo}} + \dfrac{t_{oh}}{t_{mlo}}} = \frac{1}{1 + \dfrac{\alpha/\beta_{lo}}{(\alpha/\beta_{lo})_c} + \dfrac{\beta_{hi}/\beta_{lo}}{(\beta_{hi}/\beta_{lo})_c} + \dfrac{t_{oh}}{t_{mlo}}} \qquad \text{Eq 10}$$

where the definitions of $(\alpha/\beta_{lo})_c$ and $(\beta_{hi}/\beta_{low})_c$ have been substituted in. Therefore, the effective bandwidth lies between the low bandwidth $BW_{lo}$ and a fraction of it. After optimisations, the effective bandwidth is maximised ($BW_e^{max}$), and it can be estimated by substituting the optimal hardware parameters (e.g., $FLOPS^{max}$, $BW_{lo}^{max}$, and $BW_{hi}^{max}$) in Eq 10.

## 5 Optimal implementations and XLA performance

In this section, the optimal implementations of the operations on various platforms and computers are examined. The computing speed is modelled and compared with that of the XLA implementations. The codes used to reproduce the results of the present study are available in the GitHub repository (https://github.com/xuzhen-he/XLA_numerical_models) and the pseudocodes are shown in Table 5.

On the CPU platform, parallel computing is fulfilled by compiling the source codes with the *GNU C/C++* compiler (version 9.4.0 for the PC and 10.2.1 for the HPC), which is supported by the *GCC*'s implementation of *OpenMP*. Shared-memory parallelism is achieved by simply using *OpenMP* directives on the *C/C++* loops. Executables are compiled with the highest level of optimisation (i.e., with option *-O3*) and optimised with all the instruction sets of the local machine (i.e., with option *-march = native*). Additionally, auto-vectorisation is enabled by default with this level of optimisation, the information about vectorisation is dumped with the option *-fopt-info-vec*.

On the GPU platform, parallel computing is fulfilled by compiling the source codes with the *CUDA C/C++* (version 11.7 for the PC and 11.6 for the HPC). Pseudocodes of the operations (called kernels in *CUDA*) are shown in Table 5. In *CUDA*, the kernels are invoked with grids of thread blocks, which mimics how GPU processors are physically grouped. Grids and thread blocks can be 1D, 2D or 3D. It is natural to use 1D grids and thread blocks for 1D problems and 2D grids and thread blocks for 2D problems. Within the *CUDA* kernel, the index of

**Table 5. Pseudocodes for the operations.**

|  | **CPU platform (Fulfilled with *OpenMP*)** | **GPU platform (Fulfilled with *CUDA*)** |
|---|---|---|
| **1D** | inline void substep($\mathbf{X}_1, \mathbf{X}_2,\ldots,\mathbf{X}_{Ni},\mathbf{Y}_1,\mathbf{Y}_2,\ldots,\mathbf{Y}_{No}$, $\boldsymbol{p}$) | __global__ void substep($\mathbf{X}_1, \mathbf{X}_2,\ldots,\mathbf{X}_{Ni},\mathbf{Y}_1,\mathbf{Y}_2,\ldots,\mathbf{Y}_{No}$, $\boldsymbol{p}$) |
|  | #OpenMP directive | int_type j = blockIdx.x*blockDim.x+threadIdx.x; |
|  | for (int_type j = 0; j < total_size; j++) | if (j< total_size) |
|  | #loop body to calculate $\mathbf{Y}_q[j]$, q = 1,2,...,$N_o$ | #kernel body to calculate $\mathbf{Y}_q[j]$, q = 1,2,...,$N_o$ |
| **2D** | inline void substep($\mathbf{X}_1, \mathbf{X}_2,\ldots,\mathbf{X}_{Ni},\mathbf{Y}_1,\mathbf{Y}_2,\ldots,\mathbf{Y}_{No}$, $\boldsymbol{p}$) | |
|  | #Outer-loop OpenMP directive | __global__ void substep($\mathbf{X}_1, \mathbf{X}_2,\ldots,\mathbf{X}_{Ni},\mathbf{Y}_1,\mathbf{Y}_2,\ldots,\mathbf{Y}_{No}$, $\boldsymbol{p}$) |
|  | for (int_type i = 0; i < total_size; i++) | int_type i = blockIdx.x * blockDim.x + threadIdx.x; |
|  | #Inner-loop OpenMP directive | int_type j = blockIdx.y * blockDim.y + threadIdx.y; |
|  | for (int_type i = 0; i < total_size; i++) | if (i < Nx and j < Ny) |
|  | #loop body to calculate $\mathbf{Y}_q[i,j]$ or $\mathbf{Y}_q[j,i]$, q = 1,2,..., $N_o$ | #kernel body to calculate $\mathbf{Y}_q[i,j]$, q = 1,2,...,$N_o$ |

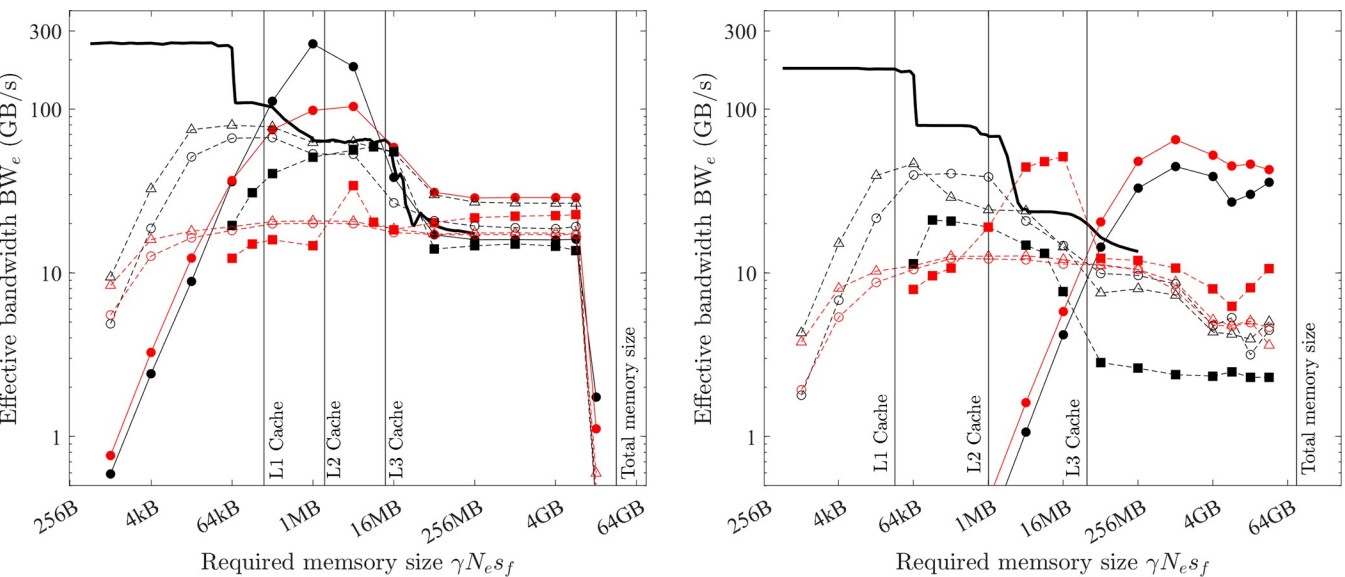

**Fig 7. Optimal implementation of vector operations on the CPU platform (double precision; solid black line = benchmarked bandwidth; lines + symbols = effective bandwidth; black = COPY1D; red = XPXPY20_1D; filled circles = *OpenMP parallel for* with the maximal no. of threads, hollow circles = *OpenMP parallel for* with one thread; hollow triangles = single-thread SIMD; filled squares = XLA).** (a) PC. (b) HPC.

the thread, the dimension of the thread block and the index of a thread block within the grid are accessed through the built-in variables *threadIdx*, *blockDim*, and *blockIdx*, respectively (Table 5).

The latency is measured by running the operations for at least 5 seconds and at least 20 times, an average latency is recorded. Array programming of the operations is achieved with the *Python* package *JAX* (version 0.3.17 for both the PC and HPC) and *JAXLIB* (version 0.3.15 for both the PC and HPC). The first run of the *JAX* implementation is not timed because it contains optimisations and compilation with XLA.

## 5.1 Element-wise vector operations: Optimal implementations on the CPU platform

For element-wise operations, all the calculations are independent and a particular input element $\mathbf{X}_l[i,\ldots]$ is only requested when calculating the corresponding output elements at the same position ($\mathbf{Y}_q[i,\ldots]$). This homogeneity makes it impossible to split the memory operations into two parts at different speeds. So, all the memory operations are at the same bandwidth with $\beta_{lo} = N_o + N_i$ and $\beta_{hi} = 0$.

On the CPU platform, two types of implementations are examined: The first is single-thread SIMD (256-bit AVX) implementation. This is achieved by using *OpenMP* directive *omp simd*. Or equivalently, without the *OpenMP* directive, the highest level of optimisation (*-O3*) will automatically vectorise (256-bit AVX) the loop. With one thread, the 256-bit AVX can achieve the highest FLOPS and bandwidth considering observations (1)~(4) of Section 3.2, and the benchmarked bandwidth from the tool *bandwidth-benchmark* is shown as black solid lines in Fig 7(A) and 7(B). The effective bandwidth of COPY1D with this implementation (black hollow triangles) is very close to the benchmarked bandwidth except for the relatively larger gap when the required memory size is small ($\gamma N_e s_f <$ L1 cache size). This gap is because the benchmark tool is written with a low-level assembly language, but the present single-thread SIMD implementation uses a high-level programming language (*C/C++*), and thus more

overhead is involved. The results also show that this overhead is negligible ($t_{oh}/t_{mlo} \ll 1$) for medium and larger problems ($\gamma N_e s_f >$ L1 cache size).

The second implementation is using the *OpenMP* directive *omp parallel for* with $N$ threads. In this method, the array is divided into $N$ sub-arrays (each is still continuous) and 256-bit AVX is used by each thread on each such sub-array. When only one thread is used, this implementation should be identical to the single-thread SIMD implementation. However, the tested effective bandwidth (hollow circles) is slightly below that (hollow triangles), which implies that using the *omp parallel for* directive involves more overhead.

When the maximal number of threads is used (12 for PC and 26 for HPC), the two CPU platforms show different features. When the problem ($\gamma N_e s_f$) is small, the overhead involved in using many threads is large (i.e., $t_{oh}/t_{mlo}$ is large), so the effective bandwidth (filled circles) is very small. Particularly, it is smaller than using a single thread (hollow triangles). The effective bandwidth with the maximal number of threads starts to surpass that of a single thread when approximately $\gamma N_e s_f >$ L1 cache size on the PC and $\gamma N_e s_f >$ L3 cache size on the HPC. The overhead cost of using multiple threads does not scale with the problem size ($\gamma N_e s_f$) but rather scales with the number of threads. So, on the HPC with more threads, a larger problem size ($\gamma N_e s_f$) is required to make the overhead cost negligible ($t_{oh}/t_{mlo} \ll 1$). Another interesting feature is that, for the PC, the effective bandwidth shows an abrupt drop for extremely large problems ($\gamma N_e s_f >$ 16 GB), which is likely because the main memory (32 GB) is made of two pieces (each has 16 GB). When $\gamma N_e s_f$ is larger than 16 GB, the 1D array is not physically continuous anymore.

The latter discussions are mainly about medium (L1 cache size $< \gamma N_e s_f <$ L3 cache size) and large problems (L3 cache size $< \gamma N_e s_f <$ 16 GB) on the PC and large problems ($\gamma N_e s_f >$ L3 cache size) on the HPC because (1) the latency of small problems is often small and their computing speed is less a concern and (2) the overhead cost for these problems is not negligible and the analysis in this study will not apply to them. So, for medium and large problems on the PC and large problems on the HPC, the optimal implementation is parallel computing with the maximum number of threads and each thread handles a continuous sub-array with SIMD instruction sets (256-bit AVX for the tested CPUs).

Element-wise operations have $\beta_{lo} = N_i + N_o$ and $\beta_{hi} = 0$. Additionally, the COPY1D operation has $\alpha = 0$. Hence, the time spent on COPY1D is only the low-bandwidth memory operations ($t_{mlo}$) and overhead cost ($t_{oh}$). However, for medium and large problems on the PC and for large problems on the HPC, the overhead cost is negligible ($t_{oh}/t_{mlo} \ll 1$), so $\zeta = 1$ in Eq 10 and $BW_e^{max} = BW_{lo}^{max}$ for COPY1D, which means that the measured effective bandwidth for the optimal implementation of COPY1D equals the maximum bandwidth. From Fig 7, for medium-size problems on the PC, the maximum bandwidth $BW_{lo}^{max}$ is variable but reaches a peak value of about 350 GB/s at $\gamma N_e s_f \approx$ 2 MB. It is constant for large problems on both the PC and HPC–about 28 and 55 GB/s, respectively. These data are summarised in Table 6. The maximum bandwidth $BW_{lo}^{max}$ is much higher for medium-size problems than that for large problems (350 vs. 28 GB/s) on the PC because the CPU platform can take advantage of the fast L1/L2/L3 caches for medium-size problems.

## 5.2 Element-wise vector operations: Optimal implementations on the GPU platform

On the GPU platform, a straightforward implementation is allocating continuous memory as arrays of floating-point numbers (f32 or f64), and each array element is handled by a GPU thread (as the pseudocode in Table 5). From observations (2)~(5) in Section 3.2, this implementation gives optimal bandwidth just without considering observation (1). The obtained

**Table 6. Hardware parameters of optimal implementations.**

| Computer | Platform | Required memory size $\gamma N_e s_f$ | $BW_{lo}^{max}$ (GB/s) | Floating-point number | $FLOPS^{max}$ (GFLOPS) | $(\alpha/\beta_{lo})_c$ | $(\beta_{hi}/\beta_{lo})_c$ |
|---|---|---|---|---|---|---|---|
| PC | CPU | Medium (1.5 ~ 12 MB) | Variable, peak of ~350 at $\gamma N_e s_f \approx 2$ MB | f64 | 140.3 | 3.2 | ~1 |
| | | | | f32 | 281.7 | | |
| | | Large (256 MB ~ 16 GB) | ~28 | f64 | 140.3 | 40.0 | ~10 |
| | | | | f32 | 281.7 | | |
| | GPU | Medium (64 kB ~ 16 MB) | Increase with $\gamma N_e s_f$, ~58.1 at $\gamma N_e s_f \approx 1$ MB | f64 | 46.6 | 6.4 | ~5 |
| | | | | f32 | 1490 | 93.2 | |
| | | Large (> 16 MB) | ~82.4 | f64 | 46.6 | 4.5 | ~5 |
| | | | | f32 | 1490 | 72.3 | |
| HPC | CPU | Large (> 38.5 MB) | ~55.0 | f64 | 487.3 | 70.0 | ~10 |
| | | | | f32 | 956.9 | | |
| | GPU | Medium (64 kB ~ 16 MB) | Increase with $\gamma N_e s_f$, ~900.0 at $\gamma N_e s_f \approx 4$ MB | f64 | 348.5 | 3.1 | ~5 |
| | | | | f32 | 11151 | 50.0 | |
| | | Large (> 16 MB) | ~372.3 | f64 | 348.5 | 7.5 | ~5 |
| | | | | f32 | 11151 | 119.8 | |

effective bandwidth of COPY1D and XPXPY20_1D with single precision is shown in Fig 8 (filled circles). When the thread block size is either 128, 256, 512 or 1024, the effective bandwidth is the same, so Fig 8 gives only the results with 512 threads.

Another implementation is to exploit the *CUDA* built-in types *float4* or *double2*. In the *CUDA* kernels, the f32 and f64 arrays are converted (*reinterpret_cast*) to arrays of *float4* and *double2*, respectively. Each GPU thread then conducts the calculation of four (f32) or two (f64) output elements. With these built-in types, the compiled program can take advantage of the instruction sets *ld.global.v4.f32* and *ld.global.v2.f64* that can read/write four f32 numbers or two f64 numbers with one instruction. The obtained effective bandwidth is shown in Fig 8 as hollow triangles.

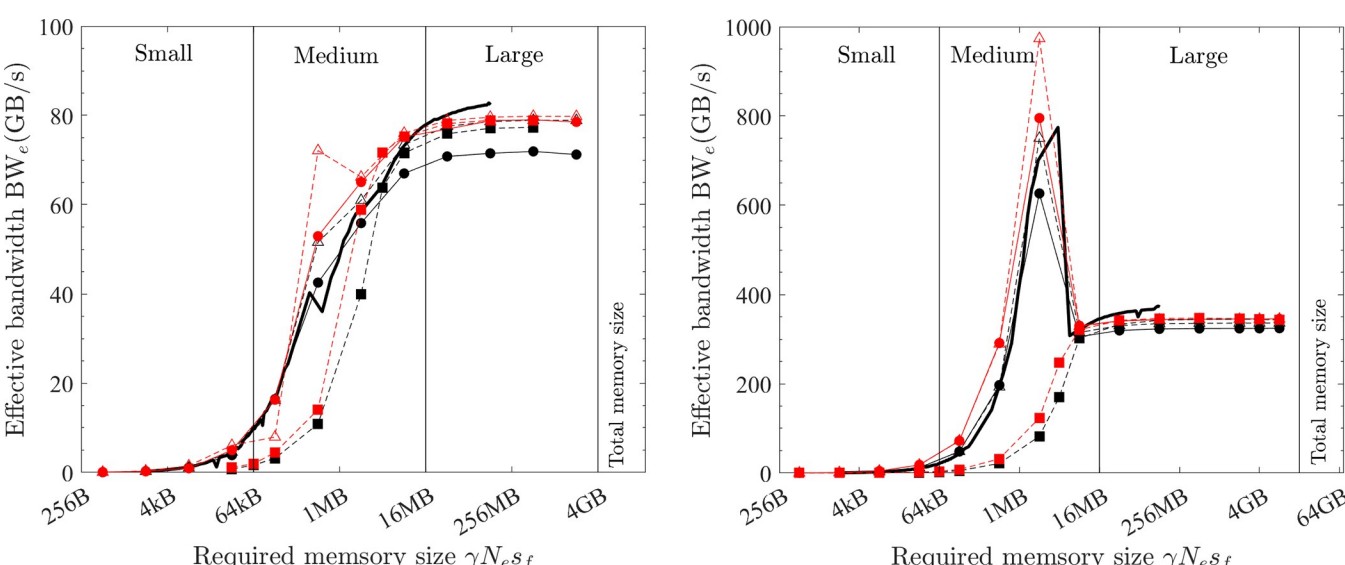

**Fig 8. Optimal implementation of vector operations on the GPU platform (single precision; solid black line = benchmarked bandwidth; lines + symbols = effective bandwidth; black = COPY1D; red = XPXPY20_1D; filled circles = *CUDA* f32, hollow triangles = *CUDA* float4; filled squares = XLA).** (a) PC. (b) HPC.

The use of these *CUDA* built-in types can give minor improvements for medium-size problems (64 kB $< \gamma N_e s_f <$ 16 MB). However, for large problems ($\gamma N_e s_f >$ 16 MB), the effective bandwidth is the same except for some minor gains for only COPY1D and SCALE1D on the PC (79 vs 72 GB/s). For other operations on the PC, both implementations give similar results (Fig 8 only shows XPXPY20_1D). Because the gain is minor, but the complexity of programming is significantly increased, the first straightforward implementation is considered optimal in this study.

Similarly, the latter discussions are mainly about medium and large problems, for which the overhead cost is negligible ($t_{oh}/t_{mlo} <<1$). So the measured effective bandwidth for the optimal implementation of COPY1D equals the maximum bandwidth (i.e., $\zeta = 1$ and $BW_e^{max} = BW_{lo}^{max}$ for COPY1D). It should be noted that the effective bandwidth of optimal COPY1D implementation is very close to the benchmarked bandwidth (black solid lines). From Fig 8, for medium-size problems (64 kB $< \gamma N_e s_f <$ 16 MB), the maximum bandwidth $BW_{lo}^{max}$ is variable and increases with the increase of $\gamma N_e s_f$. For the PC, it is about 58.1 GB/s at $\gamma N_e s_f \approx$ 1 MB. For the HPC, there is a peak of about 900 GB/s at $\gamma N_e s_f \approx$ 4 MB. For large problems ($\gamma N_e s_f >$ 16 MB), the maximum bandwidth $BW_{lo}^{max}$ is constant (about 82.4 and 372.3 GB/s for the PC and HPC, respectively).

From Fig 8, the effective bandwidth of XPXPY20_1D (red) is very close to that of COPY1D (black), which suggests that $t_{flop}/t_{mlo} << 1$ for XPXPY20_1D with single-precision calculations on the GPU platform.

## 5.3 Element-wise vector operations: Fused operations vs. simple operations

How fused operations can improve performance is illustrated with two vector operations (AXPY1D and VEC XPXPY6_1D). First consider AXPY1D. For fused operations, the scale of $\mathbf{X}[i]$ by the parameter $a$ and the addition of the result onto $\mathbf{Y}^t[i]$ is finished within one operation (i.e., in the same for-loop on CPU and in the same kernel on GPU). The AXPY1D operation can also be achieved by conducting two simple operations–in-place scale of $\mathbf{X}[i]$ first and then adding the result to $\mathbf{Y}^t[i]$ with another vector addition operation. For the fused AXPY1D, the array $\mathbf{X}$ is read once, and the array $\mathbf{Y}$ (in-place update, so for both $\mathbf{Y}^t$ and $\mathbf{Y}^{t+1}$) is read once and written once. However, if two simple operations are conducted, $\mathbf{X}$ is read twice and written once, and the array $\mathbf{Y}$ is read once and written once. So, fewer memory operations for the fused operations ($3N_e s_f$ vs. $5N_e s_f$). Fig 9 shows the measured effective bandwidth. For large problems, the effective bandwidth of fused operations (black filled circles) is ~22.8 GB/s (CPU) and ~79 GB/s (GPU). However, the simple operations have only ~14 GB/s (CPU) and ~45 GB/s (GPU)–about 60% that of fused operations and thus 67% more latency, which exactly matches the ratio of memory operations (3:5). Similar reduction of effective bandwidth is observed for medium-size problems.

VEC XPXPY6 is next examined, and results are shown as red circles in Fig 9 In this case, the ratio of memory operations is (1:6). For large problems, the effective bandwidth is 3.8 GB/s vs. 26.7 GB/s on the CPU platform, and 13.1 GB/s vs 79 GB/s on the GPU platform–the effective bandwidth of simple operations is only 16.7% of that using a fused operation and a 500% more latency. These results demonstrated that fused operations can reduce memory operations, and significantly improve the computing speed.

## 5.4 Element-wise vector operations: Modelling the optimal computing speed

Element-wise operations have $\beta_{lo} = N_i + N_o$ and $\beta_{hi} = 0$. So, for medium or large problems for which the overhead cost is negligible, the execution time is only spent on FLOPs ($t_{flop}$) and low-bandwidth memory operations ($t_{mlo}$).

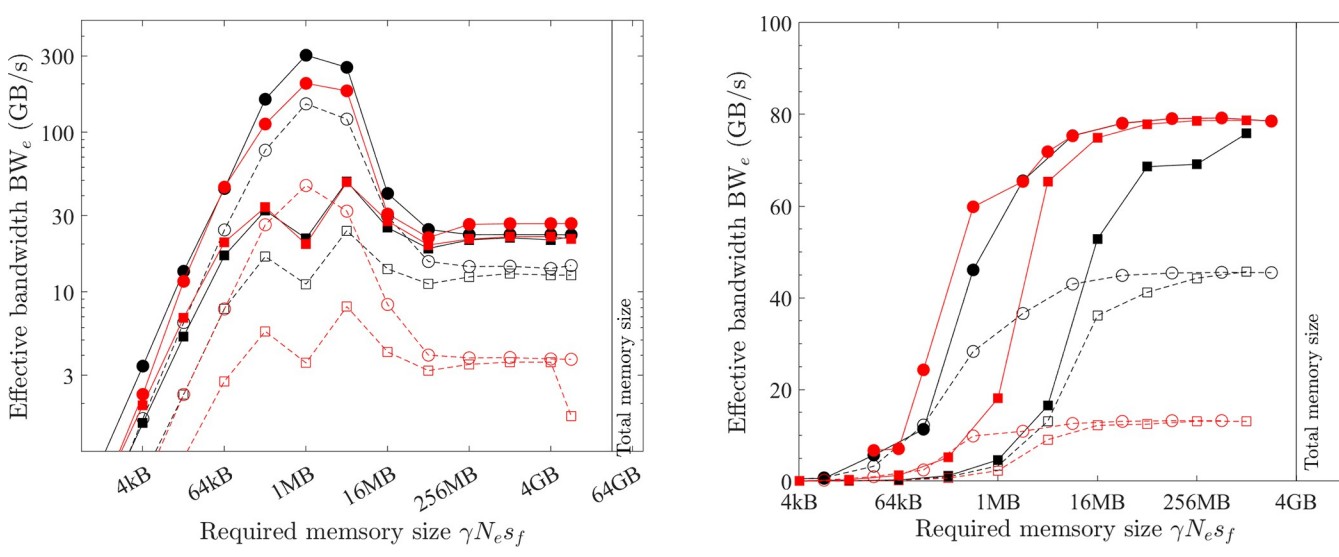

**Fig 9. Fused operations (filled circles) vs. simple operations (hollow circles) and XLA (filled squares) vs. without XLA (hollow squares) (black = AXPY1D; red = XPXPY6_1D).** (a) CPU platform of PC (f64). (b) GPU platform of PC (f64).

On the CPU platform, for large problems, the effective bandwidth of XPXPY20_1D with many FLOPs ($\alpha = 20$) is almost the same as that of COPY1D with no FLOPs ($\alpha = 0$) from Fig 7. But a visible reduction of effective bandwidth is observed for medium-size problems. Similar observations are found for f32 calculations shown in Fig 10A, in which the effective bandwidth of six operations (with various $\alpha/\beta_{lo}$) is presented.

Fig 8 shows that, with f32 calculations on the GPU platform, the effective bandwidth of XPXPY20_1D is almost the same as that of COPY_1D for both medium and large problems. Fig 10B gives results of f64 calculations, where a clear reduction of effective bandwidth is observed for XPXPY12_1D and XPXPY20_1D compared with COPY1D.

It is explained in Section 4.1 that the ratio of the time spend on FLOPs and memory operations is $t_{flop}/t_{mlo} = [\alpha/\beta_{lo}]/[(\alpha/\beta_{lo})_c]$, where $(\alpha/\beta_{lo})_c = s_f\text{FLOPS}/\text{BW}_{lo}$ is a hardware parameter

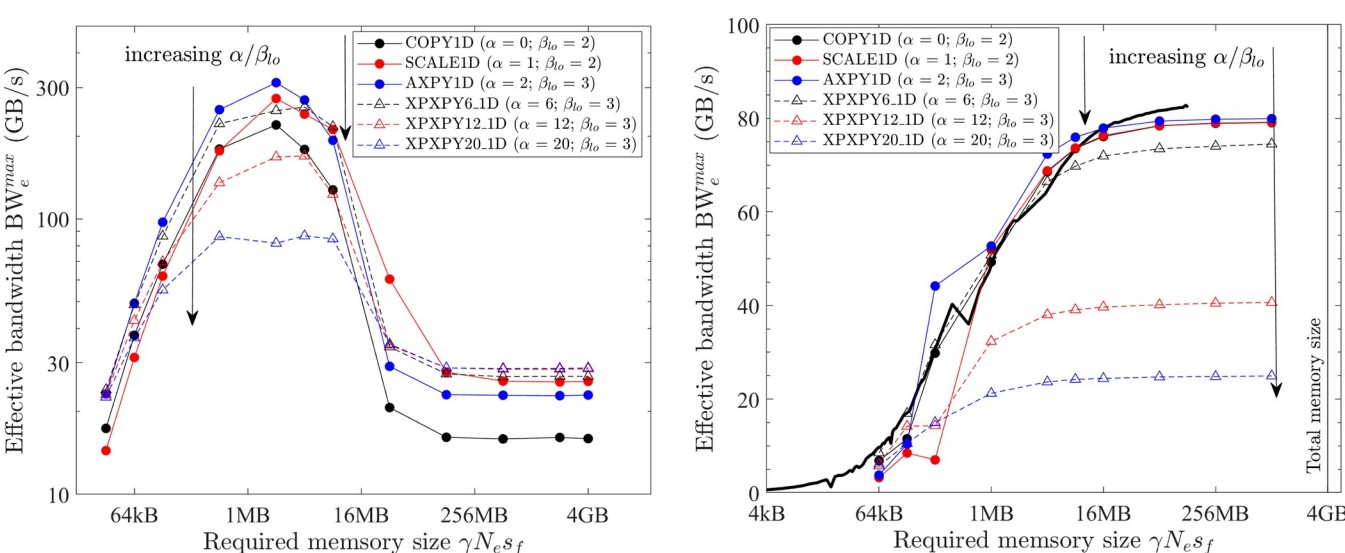

**Fig 10. Maximal effective bandwidth of various vector operations.** (a) PC CPU platform (f32). (b) PC GPU platform (f64).

measuring the relative performance of FLOPS and bandwidth. It is also a critical ratio that differentiates if an operation is memory-bound or FLOP-bound.

The critical ratio $(\alpha/\beta_{lo})_c$ is calculated for the optimal implementations and listed in Table 6. For some scenarios, the maximum bandwidth $BW_{lo}^{max}$ is not constant but variable, a typical value at a fixed $\gamma N_e s_f$ is used. For the CPU platform, the $FLOPS^{max}$ of single precision is twice that of double precision, but $s_f$ is 4 bytes for f32 and 8 bytes for f64. So, the critical ratio $(\alpha/\beta_{lo})_c$ is independent of which floating-point number is used. However, for the GPU platforms, the FLOPS of f32 calculations is significantly higher than that of f64, so the critical ratio $(\alpha/\beta_{lo})_c$ is highly sensitive to the choice of floating-point numbers.

It can be seen that $(\alpha/\beta_{lo})_c$ is very large ($> 40$) for large problems on the CPU platform and for f32 calculations on the GPU platform. So, for these cases, the time spent on FLOPs is almost negligible for the examined vector operations ($t_{flop}/t_{mlo} = [\alpha/\beta_{lo}]/[(\alpha/\beta_{lo})_c] < 20/3/40 = 0.167$). In contrast, the critical ratio $(\alpha/\beta_{lo})_c$ is relatively small (3~8) for medium-size problems on the CPU platform and for f64 calculations on the GPU platform. That is why in these cases, a clear reduction of effective bandwidth is observed for operations with larger workload ratios $\alpha/\beta_{lo}$.

The element-wise operations in this study all have $< 2$ operands. the FLOPs cannot start until all operands are read into the register, and the writing operation cannot start until the FLOPs are finished. Hence, the time spent on FLOPs and memory operations cannot overlap, and the latency is the sum of the time spent on FLOPs, memory operations, and overhead (i.e., $LT = t_{flop} + t_{mlo} + t_{oh}$). For medium or large problems where $t_{oh}/t_{mlo} << 1$, the following model regarding the effective bandwidth is then obtained.

$$BW_e = \zeta BW_{lo} = \frac{1}{1 + \frac{\alpha/\beta_{lo}}{(\alpha/\beta_{lo})_c}} BW_{lo} \qquad \text{Eq 11}$$

If the hardware parameters for optimal implementations (e.g., $FLOPS^{max}$ and $BW_{lo}^{max}$) are used, the maximum effective bandwidth $BW_e^{max}$ is obtained for Eq 11.

The relationship between the effective bandwidth and the workload ratio $\alpha/\beta_{lo}$ is shown in Fig 11. The measured maximum effective bandwidth of various element-wise operations is shown as symbols, which include tests on various platforms and computers, and with both f32 and f64 calculations. The solid lines are the model predictions by Eq 11, which agree well with the measured results. A steeper slope of the lines means a small critical ratio $(\alpha/\beta_{lo})_c$ and the lines are almost horizontal for cases with larger critical ratios, i.e., $(\alpha/\beta_{lo})_c > 40$.

The model (Eq 11) therefore suggests a useful method to roughly predict the maximum computing speed of element-wise operations, which is valid for various platforms and computers. The ratio $\alpha/\beta_{lo}$ is easily found from numerical scheme equations. The $FLOPS^{max}(s_f)$ can be estimated from hardware specifications or benchmarked. The maximum bandwidth $BW_{lo}^{max}(s_t = \gamma N_e s_f)$ can be benchmarked by running the optimal implementation of COPY1D (On the GPU platform, the result from *bandwidthTest* is the same).

## 5.5 Element-wise vector operations: The performance of XLA

With the *JAX* Just In Time (JIT) compilation, *JAX Python* functions are optimised and compiled by XLA into optimal implementations specific to the target platform (e.g., CPU or GPU platforms). If the JIT transformation is not used (without XLA), the *JAX Python* functions are executed with many built-in high-performant simple operations. With XLA, the element-wise operations expressed in array operations are fused together into an optimal operation, which is confirmed by checking the generated HLOs in the debug mode. In Fig 9, with AXPY1D and XPXPY6_1D as examples, it is shown that compared with the effective bandwidth without

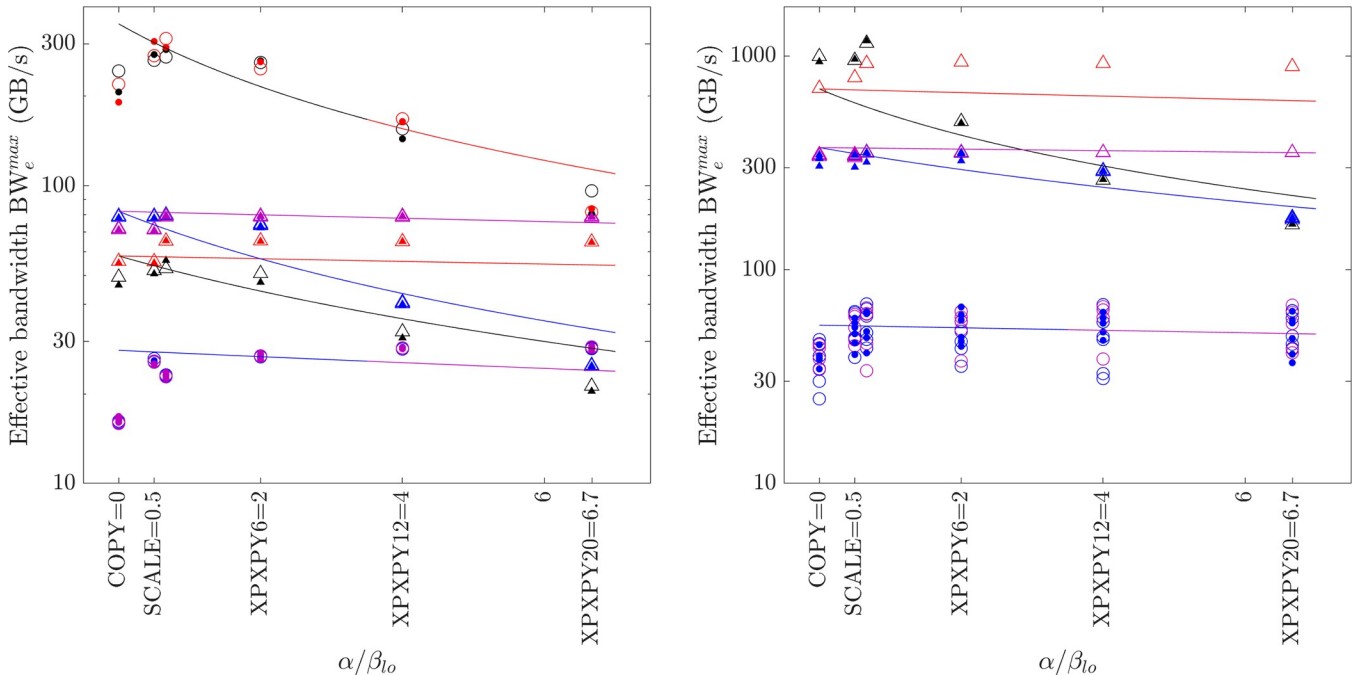

**Fig 11. Model for the effective bandwidth of element-wise operations (lines = model prediction by Eq 11; symbols = measured maximal effective bandwidth; filled symbols = matrix operations; hollow symbols = vector operations; circles = CPU; triangles = GPU; black = medium f64; red = medium f32; blue = large f64; magenta = large f32).** (a) PC. (b) HPC.

XLA (hollow squares), a significant performance gain is observed with XLA (filled squares). Additionally, the performance of XLA closely resembles the maximum performance of fused operations compiled with *C/C++* (*OpenMP or CUDA*), and the performance without XLA is very close to that of using simple operations compiled by *C/C++*. Another observation is that the effective bandwidth of *JAX Python* implementations (with or without XLA) is lower than the corresponding ones compiled with *C/C++*, which is largely due to the relatively higher overhead cost in *Python*.

The performance of XLA is examined in a stringent way by comparing it with the optimal implementations discovered in the previous subsections. The effective bandwidth of six element-wise operations (COPY, SCALE, AXPY, XPXPY6, XPXPY12, XPXPY20) is measured on various platforms and computers, and with both f64 and f32 calculations. The relative efficiency, a ratio of effective bandwidth between XLA and optimal implementations, is presented in Fig 12. Some individual results regarding the effective bandwidth of XLA are also shown in Figs 7 and 8. Some observations are:

1. On the CPU platform of the PC, the optimised HLOs are compiled with *LLVM* in an optimal way. For large problems ($\gamma N_e s_f > $ L3 cache), the effective bandwidth of XLA is only slightly below that of the optimal implementation (e.g., 22.1 vs. 28.5 GB/s for XPXPY20_1D in Fig 7A). However, for medium-size problems (L1 cache $< \gamma N_e s_f <$ L3 cache), XLA is less efficient (e.g., 34.0 vs. 90.0 GB/s for XPXPY20_1D in Fig 7A). Fig 12 shows that the relative efficiency (black symbols) is over 80% for large problems, but low (10% ~ 80%) for medium-size problems. This is likely because the overhead cost in *Python* programs is higher than that of *C/C++* programs. This cost is still relatively negligible for large problems, but will not so for medium-size problems.

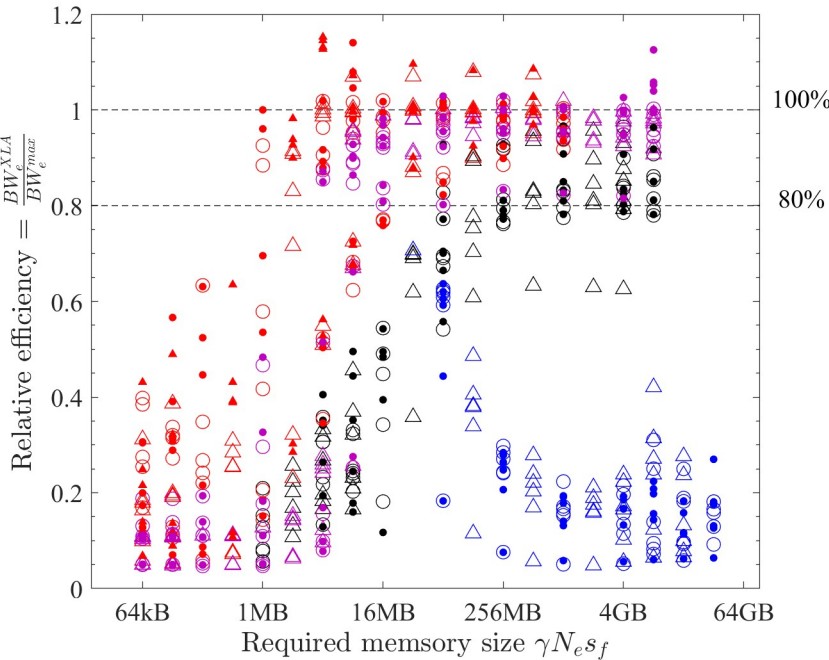

**Fig 12. Relative efficiency of XLA for element-wise operations (hollow = vector operations; filled = matrix operations; circles = f64; triangles = f32; black = PC CPU; red = PC GPU; blue = HPC CPU; magenta = HPC GPU).**

2. On the CPU platform of the HPC, the operations are successfully fused in XLA optimisation (confirmed by checking the HLOs), but the optimised HLOs are not optimally compiled with *LLVM*. Fig 7B shows that the effective bandwidth of XLA closely resembles that of single-thread SIMD implementation. Therefore, XLA performs well for small and medium problems, but is very inefficient for large problems (XLA ≈ 7 GB/s and optimal ≈ 50 GB/s for XPXPY20_1D in Fig 7B), resulting in a very low relative efficiency ($< 20\%$; blue symbols in Fig 12).

3. On the GPU platform (either PC or HPC), the performance of XLA is very close to that of optimal implementations for large problems (relative efficiency $> 90\%$; red and magenta symbols in Fig 12), but not very efficient for medium-size problems (5% ~ 90%), which is similarly due to the relatively higher overhead cost in *Python*. Some data in Fig 12 indicate a relative efficiency of over 100% on the GPU platform but not more than 120%. An examination of the generated *ptx* files (a low-level assembly language) reveals that XLA takes advantage of the instruction sets *ld.global.v4.f32* and *ld.global.v2.f64*, which can read/write four f32 numbers or two f64 numbers with one instruction. Recall that the use of these instruction sets does gain minor improvement, but is not considered optimal due to the complexity of programming.

## 5.6 Element-wise matrix operations

All the matrices tested in this study are square matrices. On the CPU platform, two nested loops are used for 2D problems. How the two loops are mapped to the two indices of matrice is critical. A row-major order is assumed, so consecutive elements in the last index are

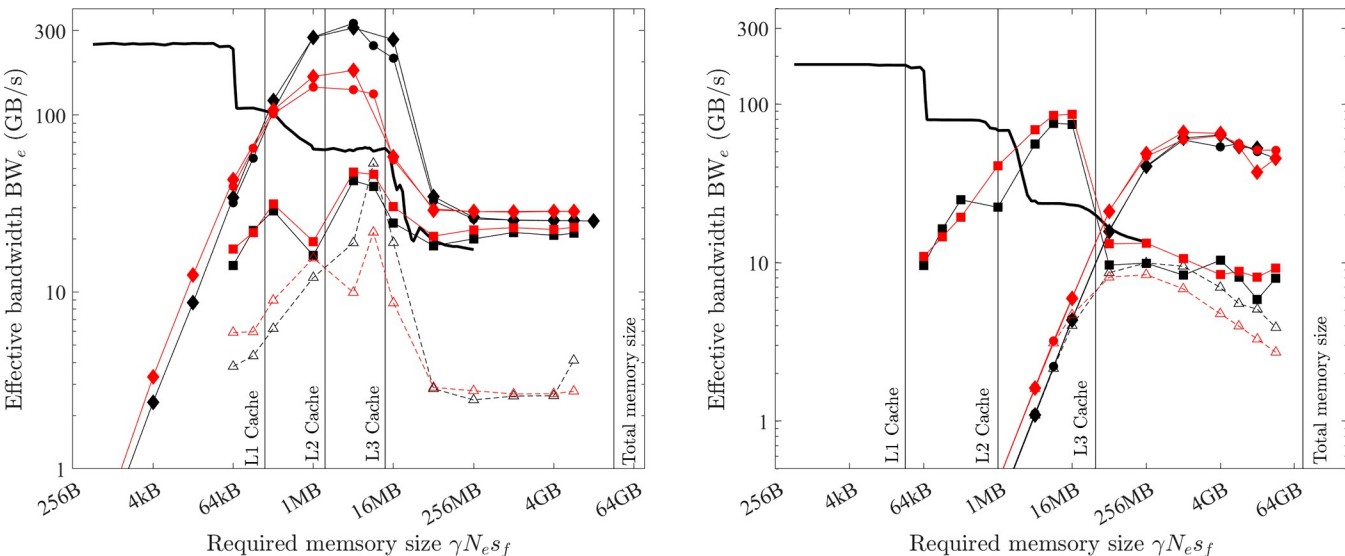

**Fig 13. Optimal implementation of matrix operations on the CPU platform (double precision; solid black line = benchmarked bandwidth; lines + symbols = effective bandwidth; black = COPY2D; red = XPXPY20_2D; filled circles = *OpenMP parallel for* with cached access; hollow triangles = *OpenMP parallel for* with uncached access; filled diamonds = optimal implementation for corresponding vector operations; filled squares = XLA).** (a) PC. (b) HPC.

contiguous in memory. If the last index is mapped to the inner loop, the first index is mapped to the outer loop, and the outer loop is decorated with the *OpenMP* directive *omp parallel for*, cached memory access is achieved. In this implementation, the array is divided into $N_x$ sub-arrays (each still continuous) and each CPU thread uses SIMD instruction sets on each such sub-array. Here $N_x$ is the size of the first index. With the maximal number of threads, the measured effective bandwidth of COPY2D and XPXPY20_2D is presented in Fig 13 as filled circles. The performance is the same as the optimal performance of the corresponding vector operations (i.e., COPY1D and XPXPY20_1D; filled diamonds in Fig 13) because these two implementations are similar, the only difference is that the array is divided by the number of threads for vector operations, but is divided by $N_x$ for matrix operations. If the loops and indices are not correctly mapped (i.e., the last index is mapped to the outer loop), the memory access pattern is uncached (Fig 6), and a significant reduction of effective bandwidth is observed (hollow triangles in Fig 13). Fig 13B shows that cached or uncached implementations are the same for small and medium problems on the HPC, which is because the overhead cost dominates for them.

On the GPU platform, *CUDA* kernels for matrix operations are invoked by 2D grids and thread blocks. The shape of the thread block ($B_x \times B_y$) affects the memory access pattern and thus effective bandwidth. Here, $B_x$ and $B_y$ are the size of the thread block for the first and last indices, respectively. When $B_x$ is 1, the memory access by GPU threads is coalesced and thus optimal. When $B_x$ is larger than 1, it is uncoalesced. Fig 14 shows that the effective bandwidth is maximised when $B_x$ is 1 (black circles in Fig 14; $B_x = 2$ also works fine), it decreases with the increase of $B_x$ (from black to red, blue, magenta, and green). Additionally, Fig 14 shows that with an optimal memory-access pattern (coalesced access for GPUs), the maximum effective bandwidth of matrix operations (black filled circles) is the same as the corresponding vector ones (filled diamonds).

The results above show that on both the CPU and GPU platforms, the maximum effective bandwidth of matrix operations is the same as the corresponding vector ones, which is also

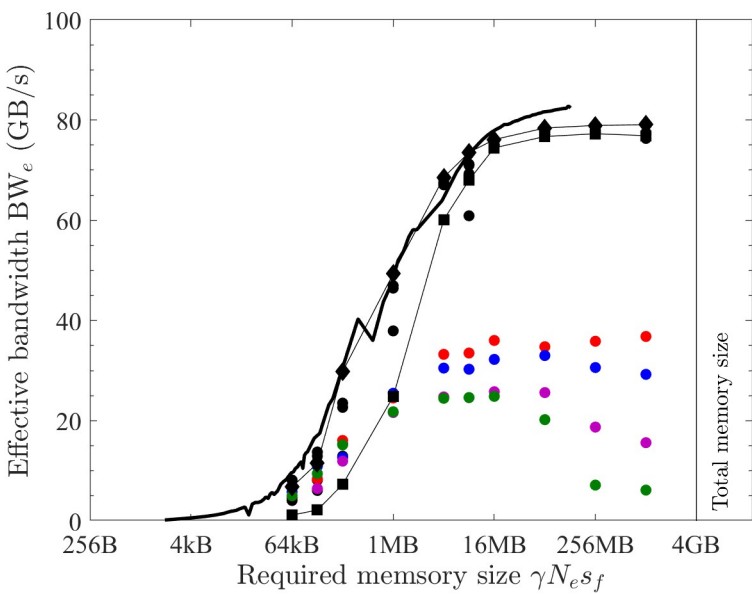

**Fig 14. Optimal implementation of COPY2D on the GPU platform of the PC (double precision; solid black line = benchmarked bandwidth; circle symbols = effective bandwidth with different shapes of thread block; black = 1 × 512, 1 × 128, 2 × 256, or 2 × 64; red = 4 × 128; blue = 16 × 32; magenta = 128 × 4; green = 512 × 1; filled diamonds + line = optimal for COPY1D; filled squares + line = XLA).**

implied by the model Eq 11 because it contains no information regarding the dimension of arrays. The data in Fig 11 also support this–filled symbols are from matrix operations and they collapse with the hollow symbols from vector operations.

The performance of XLA is shown as filled squares in Figs 13 and 14. Comparing these with the filled squares in Figs 7 and 8 which are for the corresponding vector operations, it is found that the performance of XLA for matrix operations is the same as that of corresponding vector operations–simples operations are successfully fused in XLA, and the optimised HLOs are compiled optimally for the CPU platform of the PC and also for the GPU platforms, but is compiled in a non-optimal way for the CPU platform of the HPC. Therefore, the relative efficiency is the same for both vector and matrix operations, which is supported by the data (filled symbols in Fig 12 are from matrix operations and they collapse with the hollow symbols from vector operations).

### 5.7 1D finite-difference model: Optimal implementations

The optimal implementation of the 1D model (HEAT1D) is examined in this section. On the CPU platform, two implementations are assessed: (1) Single-thread 256-bit AVX implementation, and (2) the use of *OpenMP* directive *omp parallel for* with multiple threads. Similar to vector operations, using *omp parallel for* with one thread (hollow circles in Fig 15) is almost the same as single-thread 256-bit AVX implementation (hollow triangles), but involves slightly higher overhead. Using *omp parallel for* with the maximum number of threads (filled circles in Fig 15) is optimal for medium and large problems on the PC and for large problems on the HPC.

HEAT1D and COPY1D both have $\beta_{lo} = 2$ ($\beta_{lo} = N_i + N_o$ and $N_i = N_o = 1$ for them as in Table 1). So, for a fixed mesh size ($N_e$) and floating-point accuracy ($s_f$), the workload of low-bandwidth memory operations ($\beta_{lo} N_e s_f$) is the same for them. Fig 15 shows that the effective

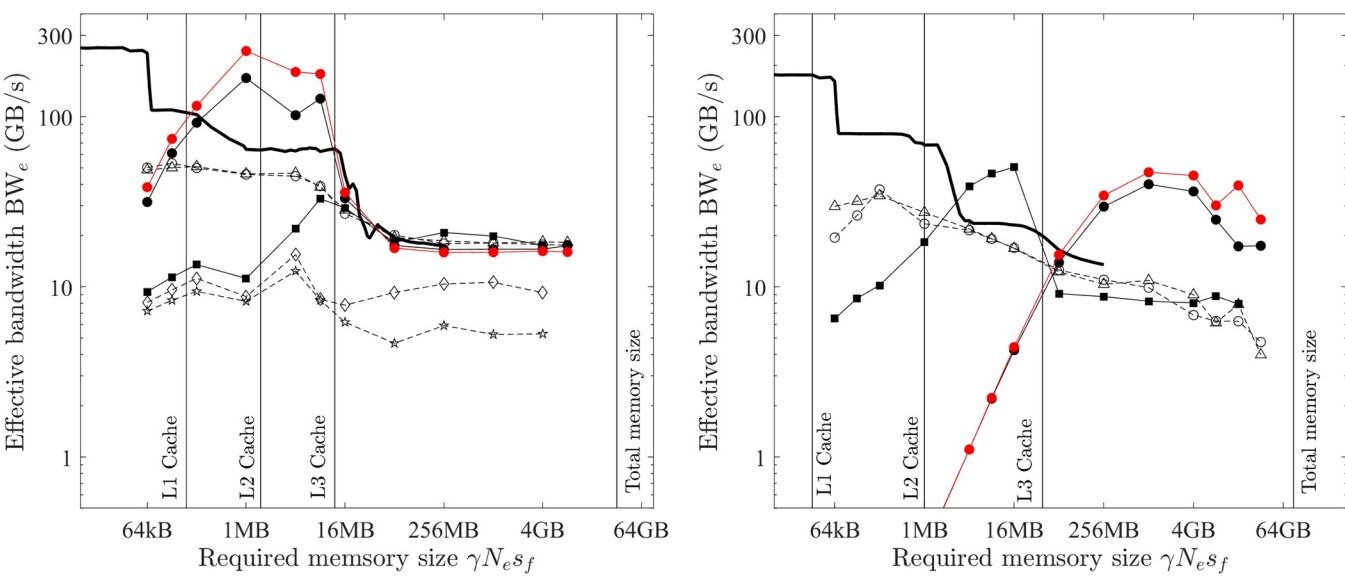

**Fig 15. Optimal implementation of HEAT1D on the CPU platform (double precision; solid black line = benchmarked bandwidth; lines + symbols = effective bandwidth; red = COPY1D; black = HEAT1D; filled circles = *OpenMP parallel for* with the maximal no. of threads, hollow circles = *OpenMP parallel for* with one thread; hollow triangles = single-thread SIMD; filled squares = XLA with slice and concatenation; hollow diamonds = XLA with convolution; hollow pentagons = XLA with roll). (a) PC. (b) HPC.**

bandwidth of the optimal implementations for them is almost the same (red filled circles vs. black filled circles), which means that the computing time for them is almost the same. However, HEAT1D involves much more FLOPs (6 vs. 0 for $\alpha$) and memory operations (4 vs. 2 for $\beta$). The reason is that the optimal implementation achieves locality of reference and uses the much faster CPU caches to conduct certain memory operations (Fig 16A). For HEAT1D, each inner element is needed to be read in the register three times. For example, $\mathbf{T}^t[i]$ are loaded three times to calculate $\mathbf{T}^{t+1}[i\text{-}1]$, $\mathbf{T}^{t+1}[i]$, and $\mathbf{T}^{t+1}[i+1]$, respectively. In the optimal implementation, these three output elements are processed by the same thread (CPU thread 1 in Fig 16A) one after another within a very short time. When the required memory size $\gamma N_e s_f$ is larger than the L3 cache size–inputs and output arrays cannot always reside on the caches, $\mathbf{T}^t[i]$ is required to be read from the main memory only once when calculating $\mathbf{T}^{t+1}[i\text{-}1]$. Afterwards, $\mathbf{T}^t[i]$ is saved on the caches and subsequent reading is from the much faster caches when calculating $\mathbf{T}^{t+1}[i]$, and $\mathbf{T}^{t+1}[i+1]$. So, the three reading operations are operated with one reading at low bandwidth from the main memory and two readings at high bandwidth from caches, which leads to $\beta_{lo} = 2$ and $\beta_{hi} = 2$. When the required memory size $\gamma N_e s_f$ is smaller than the L3 cache size–inputs and output arrays can both reside on the caches, and then the three reading are all operated at high bandwidth. But for the convenience of discussion, we still have $\beta_{lo} = 2$ and $\beta_{hi} = 2$, but rather set $\mathrm{BW_{lo}}^{max} \approx \mathrm{BW_{hi}}^{max}$.

On the GPU platform, two types of implementations are examined. In the first one, each output element is processed by a GPU thread like the pseudocode for 1D problems in Table 5. The measured effective bandwidth is shown as filled circles in Fig 17. Similarly, the effective bandwidth of HEAT1D (black filled circles) is only slightly smaller than that of COPY1D (red filled circles), and they have almost the same computing time. So, this implementation also benefits from the locality of reference (Fig 16B). $\mathbf{T}^t[i]$ is required to be loaded to the register three times to calculate $\mathbf{T}^{t+1}[i\text{-}1]$, $\mathbf{T}^{t+1}[i]$, and $\mathbf{T}^{t+1}[i+1]$, respectively. In this GPU implementation, the three output elements are processed by three GPU threads in the same thread block (i.e., $\mathbf{T}^t[i]$ is requested at the same time by three threads). So, it only needs to be read from the

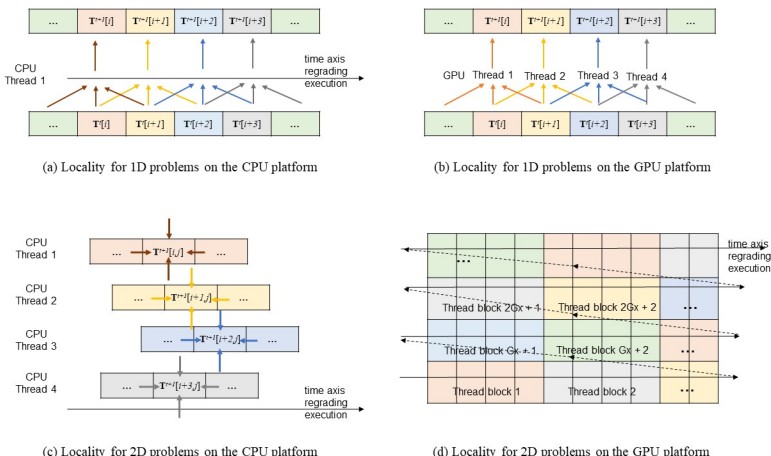

**Fig 16. Locality of reference in optimal implementations.** (a) Locality for 1D problems on the CPU platform. (b) Locality for 1D problems on the GPU platform. (c) Locality for 2D problems on the CPU platform. (d) Locality for 2D problems on the GPU platform.

GPU memory once at low bandwidth, the other two reading is from the L2 cache with high bandwidth, leading to $\beta_{lo} = 2$ and $\beta_{hi} = 2$.

Another implementation is to exploit the GPU shared memory, and the code structure is the same as the pseudocode in Table 5. Suppose the output elements processed by a thread block are from $\mathbf{T}^{t+1}[i]$ to $\mathbf{T}^{t+1}[i+blockDim-1]$, input elements with the size of $blockDim + 2$ are then read into the shared memory (i.e., from $\mathbf{T}^t[i-1]$ to $\mathbf{T}^t[i+blockDim]$), and subsequent reading of the input elements is from the shared memory, which is must faster than reading from the GPU memory (Fig 4B). With this implementation, the effective bandwidth is shown as hollow triangles in Fig 17, which is slightly smaller than that of the first one utilising the L2 cache.

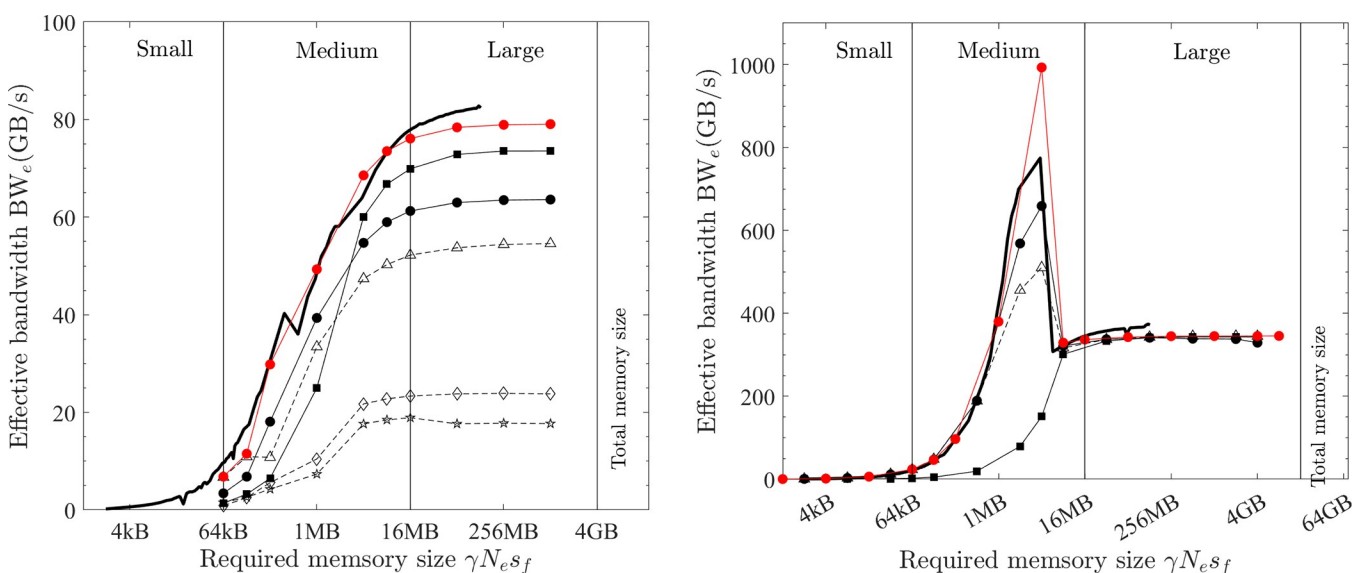

**Fig 17. Optimal implementation of HEAT1D on the GPU platform (double precision; solid black line = benchmarked bandwidth; lines + symbols = effective bandwidth; red = VEC COPY; black = HEAT1D; filled circles = *CUDA* utilising L2 cache; hollow triangles = *CUDA* utilising shared memory; filled squares = XLA with slice and concatenation; hollow diamonds = XLA with convolution; hollow pentagons = XLA with roll).** (a) PC. (b) HPC.

The reason is that some input elements staying at the boundaries of thread blocks may be read into the shared memory twice, and the high-bandwidth memory operations are also larger ($\beta_{hi}$ = 3 from the shared memory vs. $\beta_{hi}$ = 2 from the L2 cache). Therefore, the first implementation utilising the L2 cache is considered optimal. Besides, the complexity of coding is substantial when using shared memory.

## 5.8 2D finite-difference models: Optimal implementations

The optimal implementations of the two 2D models (HEAT2D and NS2D) are examined in this section. NS2D has two substeps. In the section, an overall latency (LT = $LT_{predictor}$ + $LT_{corrector}$) and an overall effective bandwidth ($BW_e$ = $15N_e s_f$/LT) of the two substeps are reported.

On the CPU platform, two implementations are assessed, which differ only by how the two indices are mapped to the two *C/C++* loops. Similar to the findings in Section 5.5 about matrix operations, the implementation with cached access (filled circles in Fig 18) is optimal and performs better than that with uncached access (hollow triangles). Comparing Fig 18 with Fig 13 shows that the effective bandwidth of the optimal implementations for HEAT2D and NS2D is only slightly lower than that of optimal COPY2D even though the two 2D models involve much more FLOPs and memory operations. Similarly, the optimal implantation benefits from the locality of references, which is illustrated in Fig 16C (with HEAT2D as an example but the same is valid for all other similar 2D models). For HEAT2D, $\mathbf{T}^t[i,j]$ are requested five times to calculate the output elements $\mathbf{T}^{t+1}[i,j]$, $\mathbf{T}^{t+1}[i,j-1]$, $\mathbf{T}^{t+1}[i,j+1]$, $\mathbf{T}^{t+1}[i+1,j]$, and $\mathbf{T}^{t+1}[i-1,j]$, respectively. The three elements $\mathbf{T}^{t+1}[i,j]$, $\mathbf{T}^{t+1}[i,j-1]$ and $\mathbf{T}^{t+1}[i,j+1]$ are processed by the same CPU thread sequentially in a very short time. The other two elements $\mathbf{T}^{t+1}[i+1,j]$ and $\mathbf{T}^{t+1}[i-1,j]$ are processed by neighbouring CPU threads. Because all threads start with the elements whose second index is 0, even though the CPU threads work independently, they will reach the three elements $\mathbf{T}^{t+1}[i,j]$, $\mathbf{T}^{t+1}[i+1,j]$ and $\mathbf{T}^{t+1}[i-1,j]$ (their second indices are all *j*) approximately the same time. Therefore, even though $\mathbf{T}^t[i,j]$ is requested five times, the five readings happen within a very short period, leading to $\beta_{lo}$ = 2 and $\beta_{hi}$ = 4.

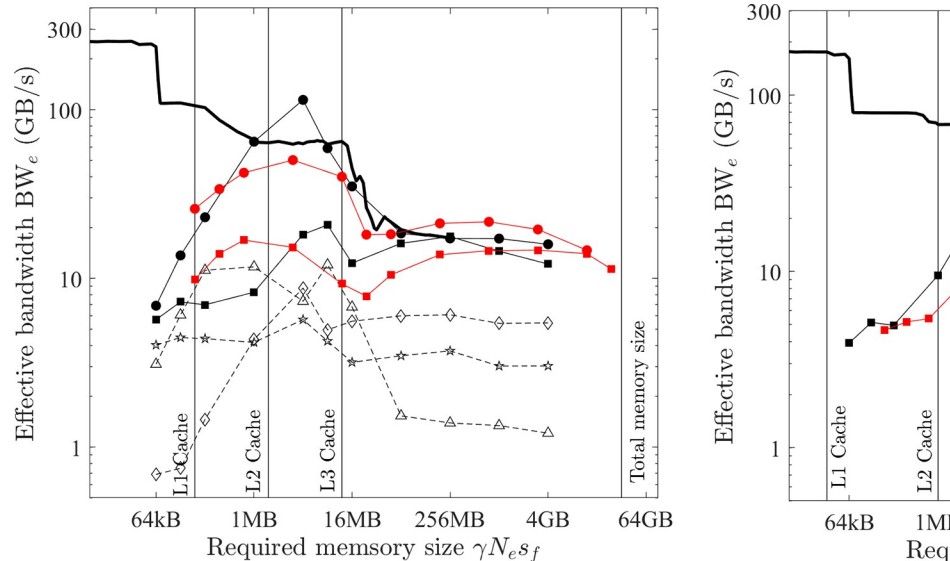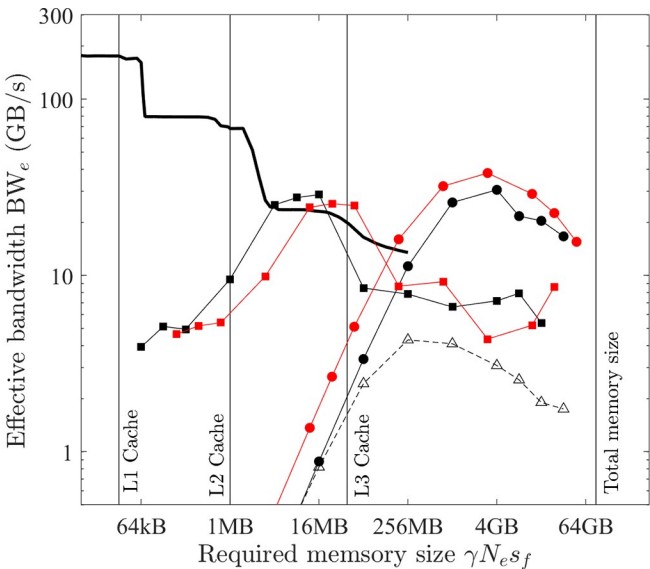

**Fig 18. Optimal implementation of HEAT2D and NS2D on the CPU platform** (double precision; solid black line = benchmarked bandwidth; lines + symbols = effective bandwidth; black = HEAT2D; red = NS2D; filled circles = *OpenMP parallel for* with cached access; hollow triangles = *OpenMP parallel for* with uncached access; filled squares = XLA with slice and concatenation; hollow diamonds = XLA with convolution; hollow pentagons = XLA with roll). (a) PC. (b) HPC.

On the GPU platform, two implementations like those for the 1D model are examined. In the first one, each output element is processed by a GPU thread, and the *CUDA* kernel is invoked with 2D grids and thread blocks ($B_x = 1$ and $B_y = 512$). The choice of $B_x = 1$ is because it will ensure the optimal memory access pattern for GPU threads (i.e., coalesced access as demonstrated in Section 5.5). The effective bandwidth is shown as filled circles in Fig 19 (black = HEAT2D and red = NS2D), which is only slightly below that of optimal COPY2D (equals the benchmarked bandwidth). This implementation also benefits from the locality of references and is demonstrated in Fig 16D. The arrows indicate the order of output elements in memory and also the execution order of thread blocks for 2D models. Similarly, each input element $\mathbf{T}^t[i,j]$ is requested five times to calculate the output elements $\mathbf{T}^{t+1}[i,j]$, $\mathbf{T}^{t+1}[i,j-1]$, $\mathbf{T}^{t+1}[i,j+1]$, $\mathbf{T}^{t+1}[i+1,j]$, and $\mathbf{T}^{t+1}[i-1,j]$, respectively. With $B_x = 1$, the three elements $\mathbf{T}^{t+1}[i,j-1]$, $\mathbf{T}^{t+1}[i,j]$ and $\mathbf{T}^{t+1}[i,j+1]$ are continuous in memory and are processed by neighbouring threads in a thread block at the same time. But the other two elements $\mathbf{T}^{t+1}[i-1,j]$ and $\mathbf{T}^{t+1}[i+1,j]$ are processed before and after the three continuous elements. After $\mathbf{T}^{t+1}[i,j]$ is processed, a whole row of output elements (number = $N_y$) needs to be processed before $\mathbf{T}^{t+1}[i+1,j]$ can be processed. So, if the L2 cache can store a whole row ($N_y$) of inputs elements, then when processing $\mathbf{T}^{t+1}[i+1,j]$, the input element $\mathbf{T}^t[i,j]$ still resides on the L2 cache and can be read into the register faster. For the largest problem tested on the GPU platforms, $N_y$ is $8 \times 1024$ for the PC and $N_y$ is $32 \times 1024$ for the HPC, which will require 64 kB (PC) and 256 kB (HPC) to store a whole row of $\mathbf{T}^t$ in f64. However, the L2 cache is much larger– 512 kB for the PC and 4 MB for the HPC. Therefore, this optimal implementation also has $\beta_{lo} = 2$ and $\beta_{hi} = 4$.

Another implementation on the GPU platform is to exploit the shared memory. For 2D problems, the size of thread blocks is $B_x = 1$ and $B_y = 512$ to have an optimal memory access pattern. Input elements with the shape of $(B_x + 2) \times (B_y + 2)$ are required to be loaded into the shared memory, and subsequent reading of the input elements is from the shared memory. Fig 19 shows that the performance of this implementation (hollow triangles) is slightly poorer than that of the first one utilising the L2 cache (filled circles), so the first one is optimal.

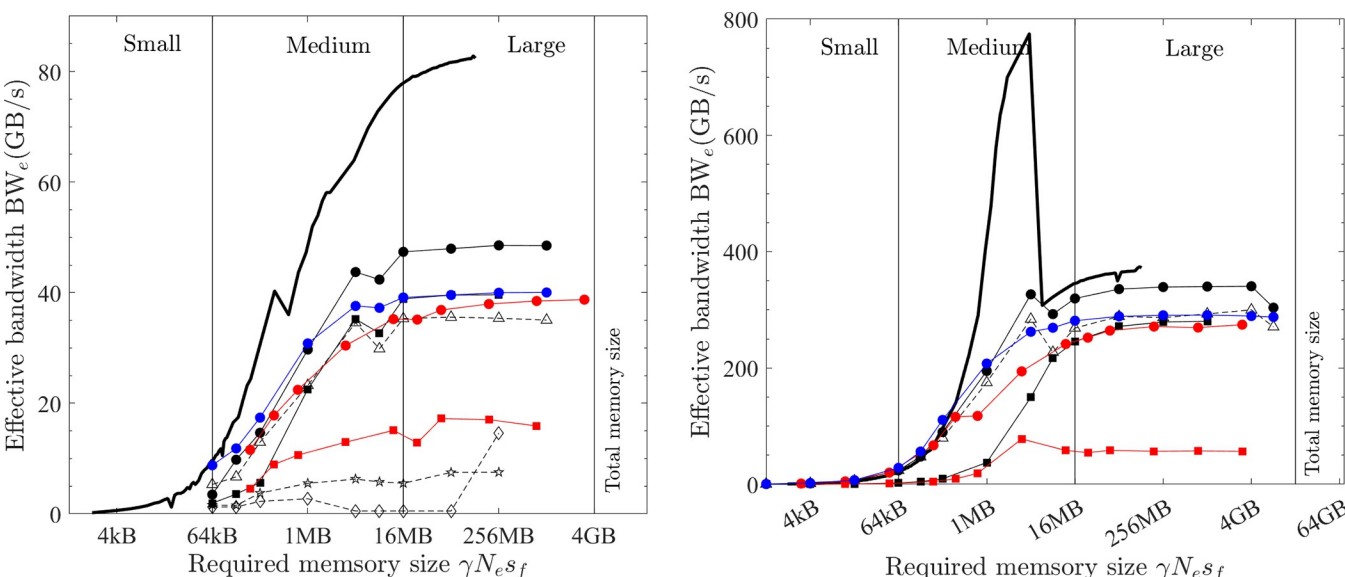

**Fig 19. Optimal implementation of HEAT2D and NS2D on the GPU platform (double precision; solid black line = benchmarked bandwidth; lines + symbols = effective bandwidth; black = HEAT2D; red = NS2D; blue = XPXPY12_2D; filled circles = *CUDA* utilising L2 cache; hollow triangles = *CUDA* utilising shared memory; filled squares = XLA with slice and concatenation; hollow diamonds = XLA with convolution; hollow pentagons = XLA with roll).** (a) PC. (b) HPC.

## 5.9 Finite-difference models: Modelling the optimal computing speed

Sections 5.6 and 5.7 show that for the optimal implementations of the numerical models on either the CPU or the GPU platform, the computing time is mostly spent on low-bandwidth memory operations and the effective bandwidth is only slightly lower than the COPY operations.

An examination of the generated computer instructions (in assembly language; *asm* for *GNU C/C++* and *ptx* for *CUDA C/C++*) reveals that FLOPs and memory operations are executed alternately for these numerical models. It is not like the element-wise operations examined in this study that FLOPs cannot start until all operands are read in the register and the writing of operation results cannot start until the calculations are done (Section 5.4). Therefore, the execution of FLOPs and memory operations may overlap each other for numerical models. Moreover, memory operations often take longer time than FLOPs, so the time spent on FLOPs is less important for numerical models. This can be demonstrated with the data. HEAT2D and XPXPY12_2D both have $\alpha/\beta_{lo}$ = 4 but HEAT2D involves more high-bandwidth memory operations ($\beta_{hi}/\beta_{lo}$ = 2 vs. 0). However, the effective bandwidth of HEAT2D (black filled circles in Fig 19) is surprisingly higher than that of XPXPY12_2D (i.e., faster). With the findings above, the latency for numerical models can be assumed to be the sum of the time spent on low- and high-bandwidth memory operations, and overhead (i.e., LT = $t_{mlo}$ + $t_{mhi}$ + $t_{oh}$). For medium or large problems where $t_{oh}/t_{mlo}$<<1, the following model is obtained.

$$BW_e = \zeta BW_{lo} = \frac{1}{1 + \frac{\beta_{hi}/\beta_{lo}}{(\beta_{hi}/\beta_{lo})_c}} BW_{lo} \qquad \text{Eq 12}$$

For large problems on the CPU platform, the low-bandwidth memory operations are from/to the main memory and the high-bandwidth memory operations are from/to caches, the critical ratio $(\beta_{hi}/\beta_{lo})_c$ = $BW_{hi}^{max}/BW_{lo}^{max}$ is therefore estimated to be about 10. For medium-size problems, $BW_{hi}^{max} \approx BW_{lo}^{max}$ (Section 5.6) and so $(\beta_{hi}/\beta_{lo})_c$ is about only 1. On the GPU platform, the low-bandwidth memory operations are from/to the GPU memory and the high-bandwidth memory operations are from/to the L2 cache, so $(\beta_{hi}/\beta_{lo})_c$ = $BW_{hi}^{max}/BW_{lo}^{max}$ is larger than 1. A value of about 5 is assumed to fit the data in Fig 20. These values of $(\beta_{hi}/\beta_{lo})_c$ are listed in Table 6.

Fig 20 shows the relationship between the effective bandwidth and the workload ratio $\beta_{hi}/\beta_{lo}$ for numerical models. The measured maximum effective bandwidth is shown as symbols, which include tests on various platforms and computers, and with both f32 and f64 calculations. The solid lines are the model predictions by Eq 12, which agree fairly with the measured results. A noticeable discrepancy is for large problems on the CPU platform, the model predictions (red lines) are always larger than the measured effective bandwidth (red symbols). The model Eq 12, therefore, suggests a useful method to roughly predict the maximum computing speed of numerical models. The ratio $\beta_{hi}/\beta_{lo}$ is easily found from numerical scheme equations, and the hardware parameters can be easily found or benchmarked.

## 5.10 Finite-difference models: The performance of XLA

It is shown in Section 2 that three methods are available for the array programming of HEAT1D and HEAT2D. With XLA, the measured effective bandwidth of the three methods is shown in Fig 15A (HEAT1D; CPU), Fig 17A (HEAT1D; GPU), Fig 18A (HEAT2D; CPU), and Fig 19A (HEAT2D; GPU). Results show that the method of using slice and concatenation (filled squares) has the highest effective bandwidth for all tests while the methods of using convolution (hollow diamonds) and roll (hollow pentagons) operations are less efficient. An

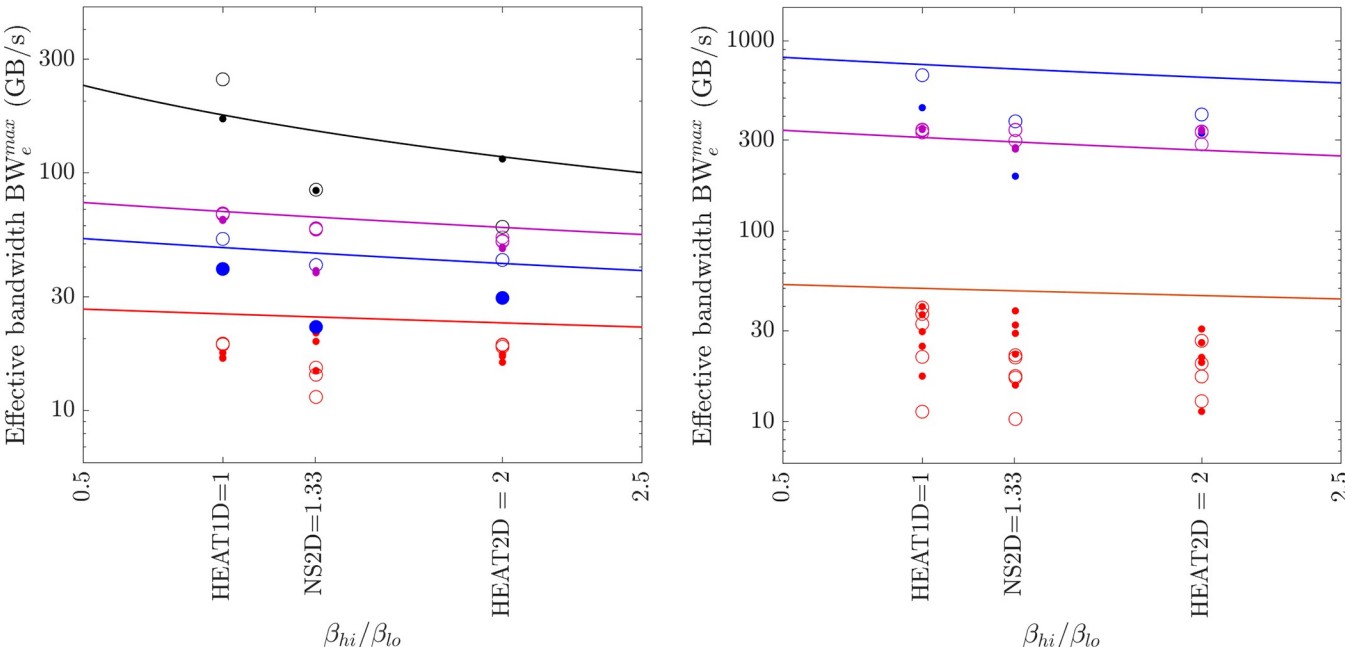

**Fig 20. Model for the maximal effective bandwidth of numerical models (lines = model prediction by Eq 12; symbols = measured maximal effective bandwidth; filled = f64; hollow = f32; black = CPU medium; red = CPU large; blue = GPU medium; magenta = GPU large).** (a) PC. (b) HPC.

examination of the generated HLOs reveals that when using slice and concatenation, the simple operations are correctly fused into one optimal operation, but the high-level operations like convolution and roll are not fused with other operations in an optimal way.

In terms of compilation with *LLVM* in XLA, it is shown that the optimised HLOs (from the slice and concatenation method) are optimally compiled for the CPU platform of the PC (Figs 15A and 18A) such that the performance of XLA (filled squares) closely resembles that of the optimal implementations (filled circles). However, on the CPU platform of the HPC (Figs 15B and 18B), optimised HLOs are compiled in a non-optimal way, and the performance of XLA (filled circles) is close to that of single-thread SIMD implementation (hollow triangles).

On the GPU platforms, the performance of XLA depends on the numerical models. For HEAT1D, the performance (filled squares in Fig 17) matches that of the optimal implementation (filled circles) and is even better than that on the PC (Fig 17A). An examination of the generated *ptx* files for HEAT1D shows that, in XLA, each GPU thread processes four (f32) or two (f64) output elements–taking advantage of the instructions *ld.global.v4.f32* and *ld.global.v2.f64*, but each GPU thread processes only one output element in the optimal implementation. For HEAT2D and NS2D, the effective bandwidth of XLA (filled squares in Fig 19) is lower than that of the optimal implementations (filled circles), and the gap for NS2D is higher than that for HEAT2D. The only difference between XLA and the optimal implantations is the number of output elements processed by a GPU thread, i.e., four (f32) or two (f64) in XLA, but one in the optimal implementation. From the generated *ptx* files for HEAT2D and NS2D, it is found that even though four (f32) or two (f64) output elements are processed by a GPU thread in XLA, the instructions *ld.global.v4.f32* and *ld.global.v2.f64* are rarely used due to the complexity of these 2D numerical models.

Fig 21 shows the relative efficiency of XLA for numerical models. The three observations (1)~(3) for element-wise operations are still valid, but one additional observation is that the relative efficiency of XLA on the GPU platform depends on the numerical model. For 1D

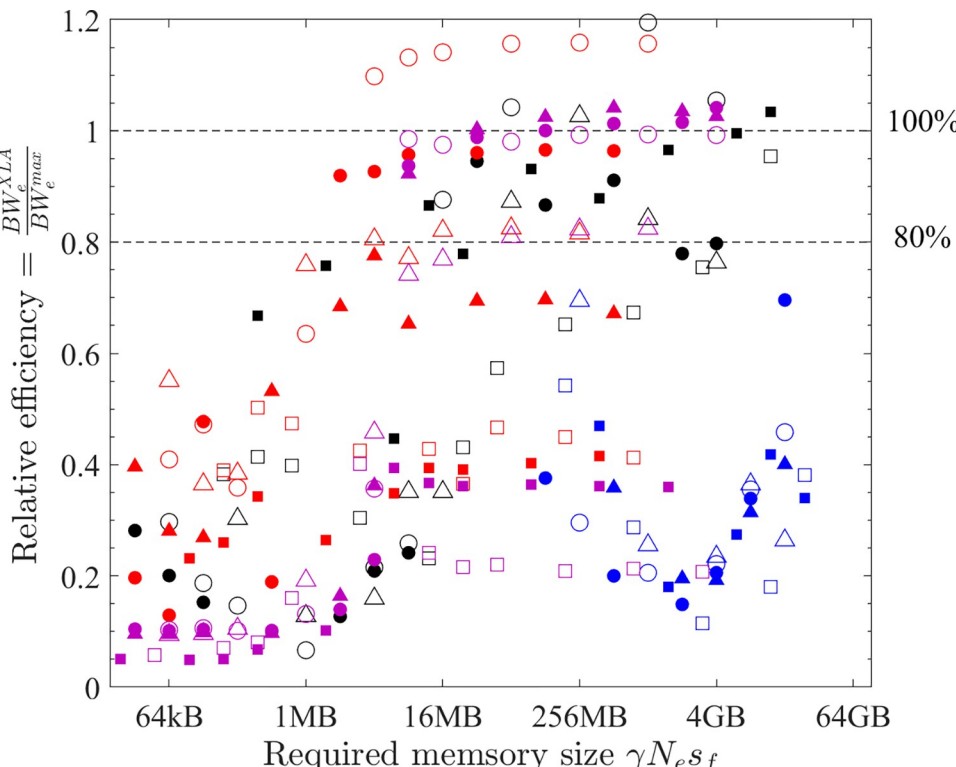

**Fig 21. Relative efficiency of XLA for numerical models (hollow = f64; filled = f32; circles = HEAT1D; triangles = HEAT2D; squares = NS2D; black = PC CPU; red = PC GPU; blue = HPC CPU; magenta = HPC GPU).**

models with a limited number of inputs and outputs (e.g., HEAT1D), it is high ($> 90\%$; and can even reach 120%) for large problems, and less efficient for medium-size problems. For 2D models, particularly the ones with many input and output arrays (NS2D), the relative efficiency is unsatisfactory even for large problems (70%~80% for HEAT2D and as low as 20~40% for NS2D).

## 6 Conclusions

This paper studies the efficiency of XLA in implementing computationally efficient numerical models. XLA is a compiler that automatically conducts optimisations (most importantly fusion to reduce memory operations) for array operations and compiles the optimised operations into target-specific programs. Speed-up is often easy to prove, this study stringently examines its efficiency by comparing the performance of XLA implementations with that of optimal implementations.

Examined models include element-wise operations (e.g., COPY, SCALE, AXPY, and XPXPYN) and numerical models (e.g., HEAT1D, HEAT2D, and NS2D). These models/operations represent a broad category of numerical models commonly encountered in the scientific computing community, hence the conclusions should easily transcend to other similar models.

Two computing platforms (backends in XLA) are examined–the shared-memory CPU platform and the shared-memory GPU platform. To obtain optimal implementations of the models on these platforms, the computing speed and its optimisation are rigorously studied by considering the different workloads and the corresponding computer performance. On the

CPU platform, optimal implementations are parallel computing with the maximum number of threads and each thread handles a continuous sub-array of output elements with SIMD instruction sets. On the GPU platform, optimal implementations are using multiple concurrent GPU threads to calculate output elements that are part of a continuous array, and each output element is processed by each such GPU thread. All optimal implementations for numerical models achieve the locality of reference such that certain memory operations are operated at high bandwidth via the caches. Two models are proposed to estimate the computing speed of element-wise operations (Eq 11) and numerical models (Eq 12) and are supported by the data.

In terms of optimisation with XLA, an examination of the generated HLOs in debug mode reveals that models expressed in low-level operations such as slice, concatenation, and array arithmetic operations are successfully fused into optimal operations, while high-level operations such as convolution and roll cannot be fused with other operations optimally.

Regarding compilation within XLA, results show that, for all examined models for the CPU platforms of certain computers (e.g., the PC), and for certain simple numerical models for the GPU platform of all computers, XLA achieves a very high efficiency ($>$ 80%) for large problems and acceptable efficiency (10%~80%) for medium-size problems–the gap is mainly due to the larger overhead cost of *Python*.

XLA obtains unsatisfactory performance for (1) all models compiled for the CPU platform of certain computers (e.g., the HPC) where the optimised operations are compiled in a non-optimal way; and (2) for high-dimensional models with many input and output arrays for the GPU platform of all computers, where XLA takes the strategy of processing 4 (single precision) or 2 (double precision) output elements with a GPU thread–hoping to use the instructions that can read/write 4 or 2 floating numbers with one instruction. However, these instructions are rarely used in the generated computer instructions due to the complexity of the models, and the performance is negatively affected. Therefore, areas for potential improvements are adding more flags to control the compilation for these non-optimal scenarios.

## Supporting information

**S1 Data.**
(XLSX)

## Author Contributions

**Conceptualization:** Xuzhen He.

**Formal analysis:** Xuzhen He.

**Funding acquisition:** Xuzhen He.

**Investigation:** Xuzhen He.

**Methodology:** Xuzhen He.

**Project administration:** Xuzhen He.

**Software:** Xuzhen He.

**Validation:** Xuzhen He.

**Visualization:** Xuzhen He.

**Writing – original draft:** Xuzhen He.

**Writing – review & editing:** Xuzhen He.

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
