## [Decision Letter · Decision Letter 0]

2 Jan 2023

PONE-D-22-33355Accelerated linear algebra compiler for computationally efficient numerical models: success and potential area of improvementPLOS ONE

Dear Dr. He,

Thank you for submitting your manuscript to PLOS ONE. After careful consideration, we feel that it has merit but does not fully meet PLOS ONE’s publication criteria as it currently stands. Therefore, we invite you to submit a revised version of the manuscript that addresses the points raised during the review process.

We look forward to receiving your revised manuscript.

Kind regards,

Viacheslav Kovtun, Dr.Sc., Ph.D.

Academic Editor

PLOS ONE

Journal Requirements:

"This work was supported by the Australian Research Council Discovery Early Career Researcher Award (DECRA; DE220100763)"

"XH was supported by the Australian Research Council (https://www.arc.gov.au/) Discovery Early Career Researcher Award (DECRA; DE220100763). The funder had no role in study design, data collection and analysis, decision to publish, or preparation of the manuscript."

Reviewers' comments:

Reviewer's Responses to Questions

**Comments to the Author**

1. Is the manuscript technically sound, and do the data support the conclusions?

Reviewer #1: Yes

Reviewer #2: Yes

2. Has the statistical analysis been performed appropriately and rigorously? 

Reviewer #1: Yes

Reviewer #2: Yes

3. Have the authors made all data underlying the findings in their manuscript fully available?

Reviewer #1: Yes

Reviewer #2: Yes

4. Is the manuscript presented in an intelligible fashion and written in standard English?

Reviewer #1: Yes

Reviewer #2: Yes

5. Review Comments to the Author

Reviewer #1: 1. in the introduction sections, author should clearly mentioned the novelty of the proposed model.

2. the quality of figures needs more improvement.

3. the comparison analysis of the proposed study with existing needs to be mentioned in both tabular and contexts.

4. the pseudo code of the proposed methodology should be provided.

Reviewer #2: Dear Authors

the manuscript titled “Accelerated linear algebra compiler for computationally efficient numerical models: success and potential area of improvement” is interesting and contains important information for readers. I can recommend this paper for publication with no modifications needed

Regards

6. PLOS authors have the option to publish the peer review history of their article (what does this mean?). If published, this will include your full peer review and any attached files.

Reviewer #1: No

Reviewer #2: No

---

## [Author Response · Author response to Decision Letter 0]

11 Jan 2023

Response to reviewers is included in a file.

---

## [Decision Letter · Decision Letter 1]

13 Feb 2023

Accelerated linear algebra compiler for computationally efficient numerical models: success and potential area of improvement

PONE-D-22-33355R1

Dear Dr. He,

We’re pleased to inform you that your manuscript has been judged scientifically suitable for publication and will be formally accepted for publication once it meets all outstanding technical requirements.

Kind regards,

Viacheslav Kovtun, Dr.Sc., Ph.D.

Academic Editor

PLOS ONE

Additional Editor Comments (optional):

Reviewers' comments:

Reviewer's Responses to Questions

**Comments to the Author**

1. If the authors have adequately addressed your comments raised in a previous round of review and you feel that this manuscript is now acceptable for publication, you may indicate that here to bypass the “Comments to the Author” section, enter your conflict of interest statement in the “Confidential to Editor” section, and submit your "Accept" recommendation.

Reviewer #2: All comments have been addressed

2. Is the manuscript technically sound, and do the data support the conclusions?

Reviewer #2: Yes

3. Has the statistical analysis been performed appropriately and rigorously? 

Reviewer #2: (No Response)

4. Have the authors made all data underlying the findings in their manuscript fully available?

Reviewer #2: (No Response)

5. Is the manuscript presented in an intelligible fashion and written in standard English?

Reviewer #2: (No Response)

6. Review Comments to the Author

Reviewer #2: (No Response)

7. PLOS authors have the option to publish the peer review history of their article (what does this mean?). If published, this will include your full peer review and any attached files.

Reviewer #2: No

---

## [Editor Report · Acceptance letter]

16 Feb 2023

PONE-D-22-33355R1 

Accelerated linear algebra compiler for computationally efficient numerical models: success and potential area of improvement 

Dear Dr. He:

I'm pleased to inform you that your manuscript has been deemed suitable for publication in PLOS ONE. Congratulations! Your manuscript is now with our production department. 

Kind regards, 

on behalf of

Professor Viacheslav Kovtun 

Academic Editor

PLOS ONE